# The intracellular inositol (pyro)phosphate receptor AtSPX1 reciprocally binds to P1BS DNA

Hayley L. Whitfield [1] ✉, Megan Gilmartin[1], Andrew M. Riley [2], Megan L. Shipton [2], Barry V. L. Potter [2], Andrew M. Hemmings [1,3] & Charles A. Brearley [1] ✉

The response of plants to phosphate starvation engages PHR (CC-MYB-PHOSPHATE STARVATION RESPONSE) transcription factors that bind to P1BS (GNATATNC) promoter elements of phosphate-starvation induced (PSI) genes. The encoded proteins include single-domain SPX (SYG1/Pho81/XPR1) proteins. SPX proteins bind PHR proteins. Current models of SPX1: PHR interaction define only a high-phosphate role for SPX1, as an inositol (pyro) phosphate-dependent negative regulator of PHR. Here, by combination of chemical synthesis, orthogonal binding assays and molecular modeling we report that full-length SPX1 binds P1BS promoter elements and inositol (pyro) phosphates with similar affinity. Inositol (pyro)phosphates and DNA are reciprocally competing ligands of SPX1. Structural models of SPX1: inositol (pyrophosphate) and of SPX1: P1BS interaction are provided beside a working hypothesis of SPX1: PHR1 interaction. The results reveal the low-phosphate function of SPX1. These findings proffer a fundamentally different perspective of SPX involvement in the phosphate starvation response (PSR).

Phosphate is an essential growth-limiting plant macronutrient. Estimated at µM level in many soils, it accumulates to mM level in plant tissues. Phytate, *myo*-inositol 1,2,3,4,5,6-hexakisphosphate ($InsP_6$), is the major form of organic phosphate in the biosphere. Both phosphate and phytate form complexes with soil minerals. This limits phosphate bioavailability to plants in most unfertilized soils. When plants sense low levels of phosphate in the rhizosphere, a PSR is initiated. This involves reprogramming of gene expression to maintain core cellular phosphate homeostasis, including upregulation of phosphate uptake, mobilization of internal storage forms and modulation of organismal transport processes[1].

A major subset of PSR genes is regulated by phosphate starvation response transcription factors (PHR), which bind to P1BS *cis*-acting promoter elements to activate expression of PSR genes[2]. A second player in the regulation of this process, SPX proteins are a diverse group of proteins critical to the regulation of PSR[3]. Crystallography of stand-alone SPX domain proteins in eukaryotes revealed high conservation of residues forming a positively charged surface capable of binding $InsP_6$ and diphosphoinositol phosphates (PP-InsPs)[4]. This affords a mechanism by which SPX proteins sense metabolites whose levels respond to the changing phosphate status of tissues.

SPX domain protein 1 (SPX1), one of four Class I (stand-alone) SPX proteins in Arabidopsis, is upregulated in Pi-deplete conditions[5]. SPX1 has been studied extensively in Arabidopsis and rice, with phosphate[6], $InsP_6$[7,8] or PP-InsP ligand-[4,8–10] dependent function emphasized. Collectively, these studies suggest that SPX1 and PHR interact only in the presence of $InsP_6$ or PP-InsP ligands, which leads to the suggestion that SPX1 is an indirect phosphate sensor. Mechanistic explanation is further provided in rice by crystallography whereby on binding of $InsP_6$ to SPX1, the 'mobile' α1 helix becomes stabilized[4]. This offers steric

[1]School of Biological Sciences, University of East Anglia, Norwich Research Park, Norwich NR4 7TJ, UK. [2]Medicinal Chemistry & Drug Discovery, Department of Pharmacology, University of Oxford, Mansfield Road, Oxford OX1 3QT, UK. [3]School of Chemistry, Pharmacy and Pharmacology, University of East Anglia, Norwich Research Park, Norwich NR4 7TJ, UK. ✉e-mail: h.whitfield@uea.ac.uk; c.brearley@uea.ac.uk

hindrance to the PHR dimer formation that is needed for PHR interaction with P1BS promoter elements[10]. In this model, the result is cessation of PSR[8]. Several of these studies report difficulties obtaining full-length protein[7,8], and consequently use C-terminal truncated protein in some experiments.

Here, we generate full-length, highly purified *Arabidopsis thaliana* SPX1 (AtSPX1). Using a suite of tools, including a chemically synthesized 2-*O*-linked InsP5 affinity matrix, we explore the binding of InsP, PP-InsP, and DNA ligands to AtSPX1, determining that AtSPX1 binds DNA only when purified to remove a bound contaminant. AtSPX1-bound DNA is displaced by inositol (pyro)phosphates, and conversely, AtSPX1-bound inositol (pyro)phosphates are displaced by DNA. These findings redefine SPX1-PHR1 involvement in PSR, whereby SPX1 binds PHR1 to disrupt binding to the P1BS promoter in the presence of inositol phosphates, but interacts with DNA itself in the absence of inositol phosphates, offering insight into a new low-phosphate function for SPX1.

## Results

### Full-length SPX1 shows little discrimination in binding affinity for InsP6, PP-InsP5 (InsP7) or [PP]2-InsP4 (InsP8) species

AtSPX1 has most commonly been studied as a truncated protein and/or as a fusion protein. We sought to investigate the binding affinities of different inositol (pyro)phosphates to full-length native AtSPX1. The chemical structures of the inositol (pyro)phosphates are shown in Fig. 1. Given the posited variation in binding pose(s) of different inositol (pyro) phosphate ligands of plant SPX[4,8–10], we initially compared the binding of two inositol phosphate fluorescence polarization probes bearing reporter substitution on the 2- and 5-positions. A newly described synthetic route for the novel molecule *myo*-inositol 5-[3-(5-fluoresceinylcarboxy)aminopropylphosphate] 1,3,4,6-tetrakisphosphate (5-FAM-InsP5) is shown in Supplementary Fig. 1. Comparison of 5-FAM-InsP5 and 2-*O*-[2-(5-fluoresceinylcarboxy)aminoethyl]-*myo*-inositol 1,3,4,5,6-pentakisphosphate (2-FAM-InsP5) as probes of AtSPX1 is shown

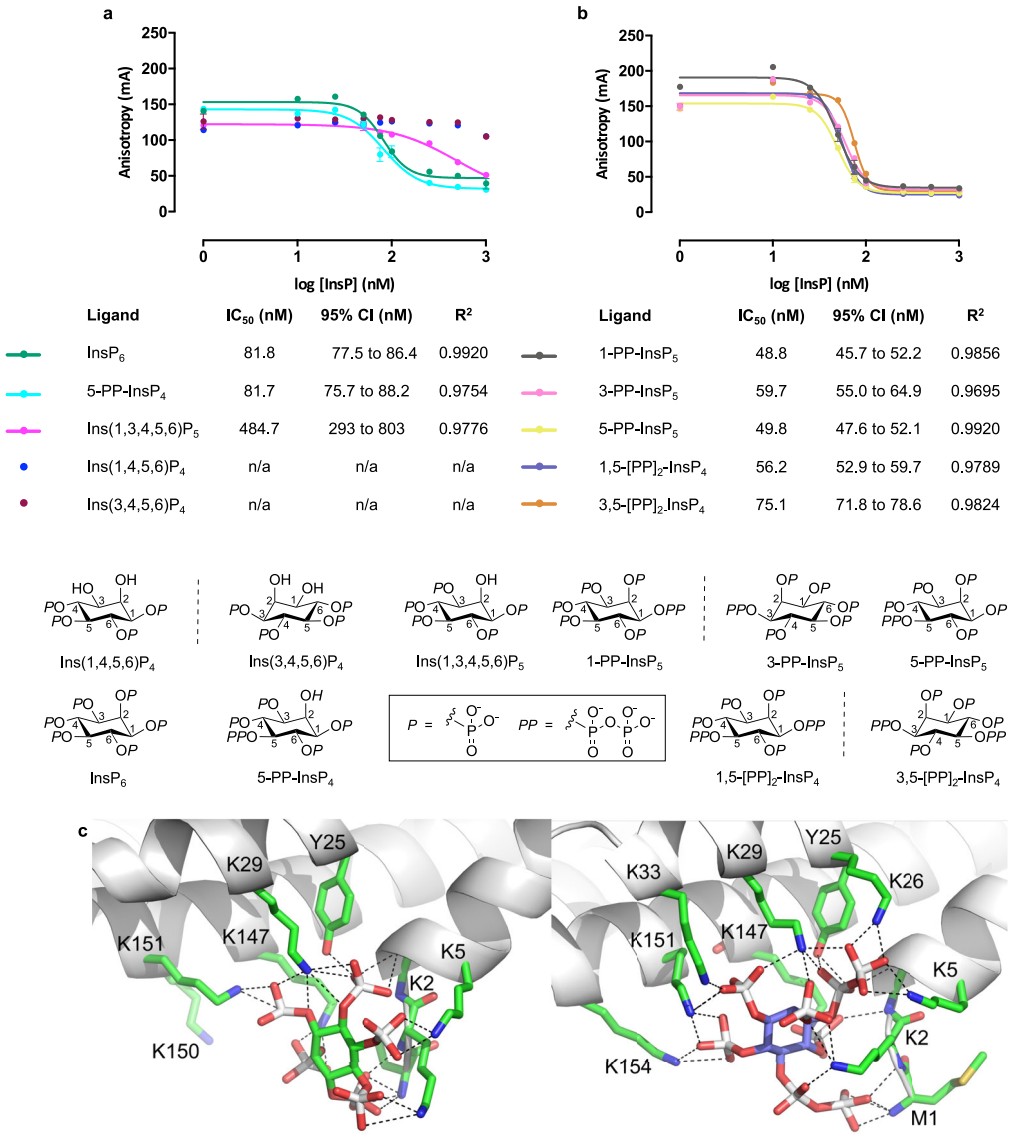

**Fig. 1 | Comparison of inositol phosphates and diphosphoinositol phosphates as ligands of SPX1.** Fluorescence anisotropy displacement assays of (full-length) AtSPX1-bound 2-FAM-InsP5 with (**a**) InsPs and 5-PP-InsP4; (**b**) PP-InsPs. Ligand structures shown. $N = 4$ replicates, repeated three times with similar results. One-way ANOVA of log IC50 revealed that the differences in IC50 were not significant at

$p < 0.05$; for InsP4 anisotropy the data could not be fitted to a 4-parameter logistic and for InsP5 did not reach saturation within this range of ligand concentrations; (**c**) side-by-side comparison of representative bound conformations of OsSPX1[1–198] with InsP6 (green) and 1,5-[PP]2-InsP4 (lilac). Note that hydrogen atoms, water molecules and counter ions have been removed for clarity.

(Supplementary Fig. 2a, chemical structures Supplementary Fig. 2b). 2-FAM–InsP$_5$ has proved useful for characterization of diverse proteins[11,12] and inositol (pyro)phosphate synthesizing enzymes[13–15] and was chosen as the most appropriate probe for AtSPX1, yielding a $K_d$ of 175 nM (Supplementary Fig. 2c, d). This allowed the use of AtSPX1 at 300 nM and 2-FAM-InsP$_5$ at 2 nM in competition (displacement) assays with a range of inositol (pyro)phosphates that are potential physiological ligands of AtSPX1 (Fig. 1a, b).

Of the inositol phosphates tested, all of which have been identified in plants[16,17], the enantiomeric pair 1D-*myo*-inositol 1,4,5,6-tetrakisphosphate, Ins(1,4,5,6)P$_4$, and 1D-*myo*-inositol 3,4,5,6-tetrakisphosphate, Ins(3,4,5,6)P$_4$, showed weak displacement of 2-FAM-InsP$_5$ at concentrations up to 1 μM (Fig. 1a). By contrast, the symmetrical (*meso*) compound *myo*-inositol 1,3,4,5,6-pentakisphosphate, Ins(1,3,4,5,6)P$_5$, a precursor of InsP$_6$ in plants[17,18], gave an IC$_{50}$ of 485 nM. With the addition of a further phosphate to the displacing ligand, InsP$_6$ (*meso*) was a stronger displacing ligand (IC$_{50}$ 82 nM) with an identical IC$_{50}$ value to 1D-5-diphospho-*myo*-inositol 1,3,4,6-tetrakisphosphate (5-PP-InsP$_4$, *meso*). The enantiomeric PP-InsP$_5$ and [PP]$_2$-InsP$_4$ molecules tested, like InsP$_6$, are substrates of *Arabidopsis thaliana* inositol tris/tetrakisphosphate kinase 1 (ITPK1) and/or *Arabidopsis thaliana* diphosphoinositol pentakisphosphate kinase 1/2 (VIH1/2)[13–15,19,20] and had similar IC$_{50}$ values (49–75 nM), showing no statistical significance between ligands, nor any difference in affinity between the respective enantiomers, viz. 1D-1-diphospho-*myo*-inositol 2,3,4,5,6-pentakisphosphate (1-PP-InsP$_5$) and 1D-3-diphospho-*myo*-inositol 1,2,4,5,6-pentakisphosphate (3-PP-InsP$_5$) or 1D-1,5-bis-diphospho-*myo*-inositol 2,3,4,6-tetrakisphosphate (1,5-[PP]$_2$-InsP$_4$) and 1D-3,5-bis-diphospho-*myo*-inositol 1,2,4,6-tetrakisphosphate (3,5-[PP]$_2$-InsP$_4$) (Fig. 1b). In this respect, the similarity of ligand affinity is much like that shown by the SPX2 domain of Vacuolar transporter chaperone complex subunit 2 (Vtc2)[21]. In summary, full-length AtSPX1 shows little enantiomeric preference and cannot, on its own, distinguish PP-InsPs from InsP$_6$, the much more abundant form of InsP in a cellular environment.

It is important to note that the available structure of OsSPX1 is that of a truncated form fused at its C-terminus with bacteriophage T4 lysozyme (SPX1$^{1–198\text{-T4-lysozyme}}$), hereafter SPX1 PDB: 7E40, whereas the gel filtration and isothermal titration calorimetry (ITC) experiments accompanying the structure were performed with full-length protein (SPX1$^{1–259}$)[8]. Therefore, in the case of both SPX1 PDB: 7E40 and truncated SPX2 (mH2A1.1$^{181–366}$-tagged SPX2$^{1–202/Δ47-59}$/InsP$_6$/PHR2$^{225–362}$), hereafter SPX2 PDB: 7D3Y[22], there are c. 60-80 amino acids of the SPX1/2 protein(s) unaccounted for in the published works. We were also unable to crystallize full-length AtSPX1. Modeling software (both AlphaFold[23] and RoseTTAFold[24]) places this additional amino acid sequence prominently across the SPX1: PHR2 contact surface residues of the Zhou et al.[8] model (Supplementary Fig. 3). It seems likely, therefore, that this unstudied C terminal region could influence SPX: PHR interaction and must be a consideration when interpreting previously published OsSPX crystal structures or modeling described herein.

We performed induced fit docking (IFD) because neither the regiochemical composition of plant PP-InsP$_5$ and [PP]$_2$-InsP$_4$, nor the crystallographic pose of PP-InsPs to plant SPX proteins, is known. We chose to use the truncated structure (SPX1 PDB: 7E40) as a receptor, since conservation between AtSPX1 and OsSPX1 is high in the InsP binding region, and a structure offers a more accurate starting point than a model of AtSPX1, such as that from AlphaFold. Predictions of the position of the deleted C-terminal region did not appear to interfere with the InsP positive binding patch; thus, modeling with the more accurate truncated form was considered appropriate. Induced fit modeling revealed a similar pattern of ligand binding preference to that of the polarization assays, with 1,5-[PP]$_2$-InsP$_4$, 3,5-[PP]$_2$-InsP$_4$ and 5-diphospho-*myo*-

inositol 1,2,3,4,6-pentakisphosphate (5-PP-InsP$_5$) showing the lowest IFD score (and thus by implication highest binding affinity) and with InsP$_6$ somewhat (ca. 14 kcal mol$^{-1}$) higher (Supplementary Table 1). The highest scoring poses from IFD were subsequently subjected to molecular dynamics simulations, and ligand binding free energies ($\Delta G$) were estimated by the MM/PBSA (molecular mechanics/Poisson–Boltzmann surface area) approach (Supplementary Methods). The $\Delta G$ values obtained (Supplementary Table 2) showed a positive but weak correlation with IC$_{50}$ (Supplementary Fig. 4a). In addition, computational scanning alanine mutagenesis allowed a prediction of the roles of individual SPX1 binding site residues in ligand recognition (Supplementary Fig. 4b, Supplementary Table 3): the values represent estimates of the change in $\Delta G$ ($\Delta\Delta G$) on substitution of individual residues with an alanine residue. Finally, representative binding poses were extracted from the resulting trajectories (Supplementary Fig. 5). These were taken from the most highly populated clusters of similar structures of each ligand and binding site residues observed during the ten 5 ns molecular dynamics trajectories used to calculate $\Delta G$ (Supplementary Figs. 6–13). The representative poses of ligands were similar, but with the 'planes' formed by the carbon atoms of the inositol rings[25] displaced relative to each other and, in some cases, rotated. This appears to arise from differential interactions of phosphate and pyrophosphate groups with the protonated N-terminus of the protein and charged sidechains of the numerous lysine residues in this region. The pose predicted for InsP$_6$ differs somewhat from that observed in the X-ray crystal structure (PDB: 7E40) and is likely due either to lattice contacts involving InsP$_6$ in the crystal structure of its complex with OsSPX1 (PDB: 7E40) and/or to the involvement of residues such as serine 0 (S0) present in the OsSPX1 crystal structure as a cloning artifact.

Docking of 1,5-[PP]$_2$-InsP$_4$ results in the sidechains of residues K2, K26, K29, K33, K147, K151, and K154 forming electrostatic contacts with the ligand, whereas for InsP$_6$ only residues K2, K5, K29, K147, and K151 do so. The net result is a total of 17 polar interactions formed by this diphosphoinositol phosphate and only ten by InsP$_6$. In addition, only 12 polar contacts are made by 5-PP-InsP$_4$ with K2, K5, K29, K147, K150, and K151, suggesting increased affinity with overall negative charge represents a possible basis for the variation in IC$_{50}$ values measured for these ligands. Scanning alanine mutagenesis suggests a strong influence of residues K2, Y25, K29, and K151 in determining the affinity of ligand binding (Supplementary Table 3). Additionally, for 1,5-[PP]$_2$-InsP$_4$, the tail of the α1 helix (including the N-terminal residue M1) rotates toward the ligand, allowing formation of polar contacts with the presumably protonated SPX1 N-terminus. A similar situation arises with other ligands, such as 3,5-[PP]$_2$-InsP$_4$ (Supplementary Fig. 5b, c). Free energy landscape analysis was performed for each of the ligands bound to OsSPX1 and for the uncomplexed protein (Supplementary Fig. 14). These plots suggest that InsP$_6$ and diphosphoinositol phosphate binding to SPX1 can alter the low-frequency molecular motions of the protein. This is unsurprising, and other authors have noted that ligand binding can remodel the conformational space of a protein, shifting energy minima and stabilizing specific states[26]. In summary, the conformations of several SPX1 residue sidechains are dependent on the identity of the bound ligand, whereby each ligand appears to cause differential modulation of the electropositive nature of the binding surface (Supplementary Fig. 5). Pipercevic et al.[21] have shown that, while InsP$_6$ has the most pronounced effect on the global thermal stability of SPX2 of Vtc2, PP-InsP ligands allow motions of the α7 helix that is constrained by InsP$_6$.

## AtSPX1 binds the P1BS element

Others have noted difficulties obtaining soluble full-length AtSPX1 or OsSPX2 proteins[7,8,22], with authors using HA-, GST-, or MBP-tagged

and/or truncated proteins. Consequently, biophysical analysis of full-length SPX (in isolation) is limited, and that of truncations is qualitative rather than quantitative in nature[22]. Given the roles posited for the N-terminal α1 helix in modulating PHR binding[8,22], we sought to generate a stable, tag-free protein using a cleavable N-terminal Histidine (His) tag. In optimization of solubility, it was found that AtSPX1 was more stable in high salt buffers. We noted that initial AtSPX1 protein preparations had a high $A_{260: 280}$, suggesting the presence of a contaminant. Two preparations of AtSPX1 were purified side-by-side using Ni NTA and Heparin columns (Prep A) or with a high salt wash step prior to these purification steps (Prep B), generating proteins with $A_{260: 280}$ of 1.24 and 0.76, respectively. These two purified stocks of AtSPX1 were subjected to fluorescence polarization with 2-FAM-InsP$_5$, and although both could bind, the stock with higher $A_{260: 280}$ showed lower affinity for the probe (Supplementary Fig. 15a). Given the possibility that the contaminant was DNA, a 5′–12mer FAM-DNA probe, hereafter FAM–DNA, was used to determine whether AtSPX1 binds DNA. All DNA probe sequences used throughout are described in Supplementary Table 4. Indeed, although AtSPX1 Prep A did not bind the FAM-DNA probe, Prep B did (Supplementary Fig. 15b). Separately, we used Human Src Homology 2 Domain-Containing Inositol 5-Phosphatase 2 (HsSHIP2) as a control protein. This well-characterized 5-phosphatase[27] has, like SPX1, a substantial positive patch on the surface[12]. Despite HsSHIP2 binding 2-FAM-InsP$_5$ with $IC_{50}$ ($\approx K_d$) of 121 nM[12], it did not bind the FAM-DNA probe (Supplementary Fig. 15c). AtSPX1 interaction with DNA was confirmed by native acrylamide gel electrophoresis (Supplementary Fig. 15d), where interaction of AtSPX1 prep B with FAM-DNA probe was visualized, but minimal interaction was observed between AtSPX1 prep A and the DNA probe. The InsP kinase *Arabidopsis thaliana* inositol tris/tetrakisphosphate kinase 4 (AtITPK4)[14] was also included in this binding assay as an InsP binding control. It showed no interaction with the DNA probe at an equal concentration to AtSPX1. A comparison of 2-FAM-InsP$_5$ and 5′-FAM-P1BS binding to AtSPX1 was performed under identical conditions, yielding $K_d$ of 175 nM (Supplementary Fig. 2d) and 630 nM (Supplementary Fig. 15e), respectively. Given that SPX1 is a nuclear-localized protein, the relatively similar (within 3–4-fold) binding affinities of FAM-InsP$_5$ and FAM–DNA suggest that DNA is accessible to this protein. Indeed, weaker binding affinities underpin current mechanistic explanations of PP-InsP action[4,5,8].

AtSPX1 was tested for affinity towards different FAM–DNA probes of P1BS: single copy (P1BS), randomized sequence of a single copy (P1BSr), P1BS repeat (4×P1BS) and P1BS randomized sequence of the P1BS repeat (4×P1BSr) using anisotropy (Fig. 2a). One way ANOVA of $logIC_{50}$ values showed that AtSPX1 had significantly ($P < 0.001$) weaker affinity towards the P1BSr than P1BS probe. The same was true ($P < 0.05$) of 4×P1BSr and 4×P1BS. The relatively small difference in binding affinities for single repeat DNA probes of the same length and AT/GC content, but with different base-order, suggests that AtSPX: DNA interaction has some sequence-specific contributions. The salt (NaCl) dependence of binding of FAM-P1BS, more pronounced than for 2-FAM-InsP$_5$ (Supplementary Fig. 15f), suggests that electrostatic contacts are formed between SPX1 and DNA. Comparison of single probe (P1BS) to repeat probe (4×P1BS) $logIC_{50}$ revealed significantly weaker binding ($p < 0.0001$) of AtSPX1 toward the repeating P1BS probe, suggesting that P1BS repeat length also plays a role in AtSPX1 binding site specificity.

Data were converted to a fraction of probe bound and subjected to a one-site total binding fit to give comparative $K_d$ values (Fig. 2b). All probes showed binding to AtSPX1, with the strongest affinity for FAM-P1BS ($K_d$ 174 nM), with the randomized sequence showing slightly weaker binding ($K_d$ 244 nM). The 4×P1BS probe had a lower affinity for AtSPX1 ($K_d$ 474 nM for 4×P1BS and $K_d$ 690 nM for 4×P1BSr).

The interaction between AtSPX1 and DNA was further analyzed by EMSA, where a mixed ssDNA and dsDNA pool of a single concentration was incubated with an increasing concentration of AtSPX1. AtSPX1 bound both ssDNA and dsDNA (Fig. 2c), whereas AtITPK4 did not.

The deep learning guided modeling software RoseTTAFoldNA[24] generated models of SPX: DNA binding, using 4xP1BS sequence with OsSPX1$^{1-198}$ (Fig. 2d), OsSPX1 full-length, AtSPX1$^{1-198}$, or AtSPX1 full-length (Supplementary Fig. 16). All models proposed showed binding of the DNA across the positively charged surface, in which the inositol phosphate binding site is found, close to the α1 helix. Modeling with the full-length proteins showed the same binding surface as the truncated protein, albeit with the DNA rotated at a different angle (Supplementary Fig. 16), suggesting this additional C-terminal region would not alter this binding position. Model outputs from AlphaFold 3[23] predicted the same binding site.

## AtSPX1 immobilized on inositol phosphate resin is displaced by DNA and InsP$_6$

The discovery of DNA-binding function makes the interplay between AtSPX1 binding of DNA and inositol (pyro)phosphates a key question when considering the function of SPX1. We employed pull-down assays to investigate this further. For this, an InsP$_5$ affinity matrix with the inositol ring coupled via the axial 2-oxygen atom (Supplementary Methods and Supplementary Fig. 17) was incubated with AtSPX1. The resin was washed with buffer or with increasing concentration of DNA or InsP$_6$ in buffer. Samples analyzed by sodium dodecyl sulfate-polyacrylamide gel electrophoresis (SDS-PAGE) showed a progressive displacement of AtSPX1 from the matrix as either InsP$_6$ or DNA was increased, but not with the addition of buffer alone (Fig. 3a). On the final addition of buffer, some protein was displaced, likely due to the prolonged assay period.

## AtSPX1-bound inositol phosphate is displaced by DNA

In an orthogonal assay, we observed the displacement of 2-FAM-InsP$_5$ from AtSPX1 by the addition of DNA. An $IC_{50}$ of 2.57 μM (Fig. 3b) was estimated. The approximate DNA concentration at which displacement occurs in the matrix pull-down assay (Fig. 3a, middle) and anisotropy assay are similar. To exclude the possibility that Pi contamination in the DNA could result in an effect on the AtSPX1 protein binding, rather than the DNA itself, suppressed ion conductivity high-pressure liquid chromatography (HPLC) was used to determine Pi content of the DNA. The potential interference by Pi was excluded (Supplementary Table 5), with concentration falling substantially below that employed in studies of SPX1: PHR1 interactions[4,6] and Pi concentration in plants[28]. Both AlphaFold 3 and RoseTTAFoldNA modeling of SPX1: DNA (the latter shown in Fig. 3c) predicted that the binding site of DNA overlaps with that of InsPs, validated here by displacement of InsP from SPX1 with an increase in DNA (Fig. 3a, b).

## P1BS-immobilized AtSPX1 is displaced by inositol phosphates or free P1BS/GC-rich DNA probe

With the knowledge that AtSPX1-bound InsPs are displaced by DNA, the reverse was investigated. Biotin-tagged P1BS dsDNA, bound to Streptavidin Sepharose resin, was incubated with AtSPX1, washed, then boiled in SDS buffer which showed AtSPX1 had bound to the P1BS DNA target but not to control resin (Supplementary Fig. 18a). Initial tests with a single concentration of InsP$_6$ or dsDNA showed displacement of AtSPX1 from the P1BS dsDNA-bound resin (Supplementary Fig. 18a, b). To investigate binding affinity, the P1BS DNA-bound matrix was incubated with AtSPX1, washed and increasing concentration of DNA or InsP$_6$ or buffer were applied (Fig. 4a). Displacement of AtSPX1 from the P1BS matrix occurs with the addition of either InsP$_6$ (Fig. 4a, top) or DNA (Fig. 4a, middle) above the background control (buffer addition, Fig. 4a, bottom) samples.

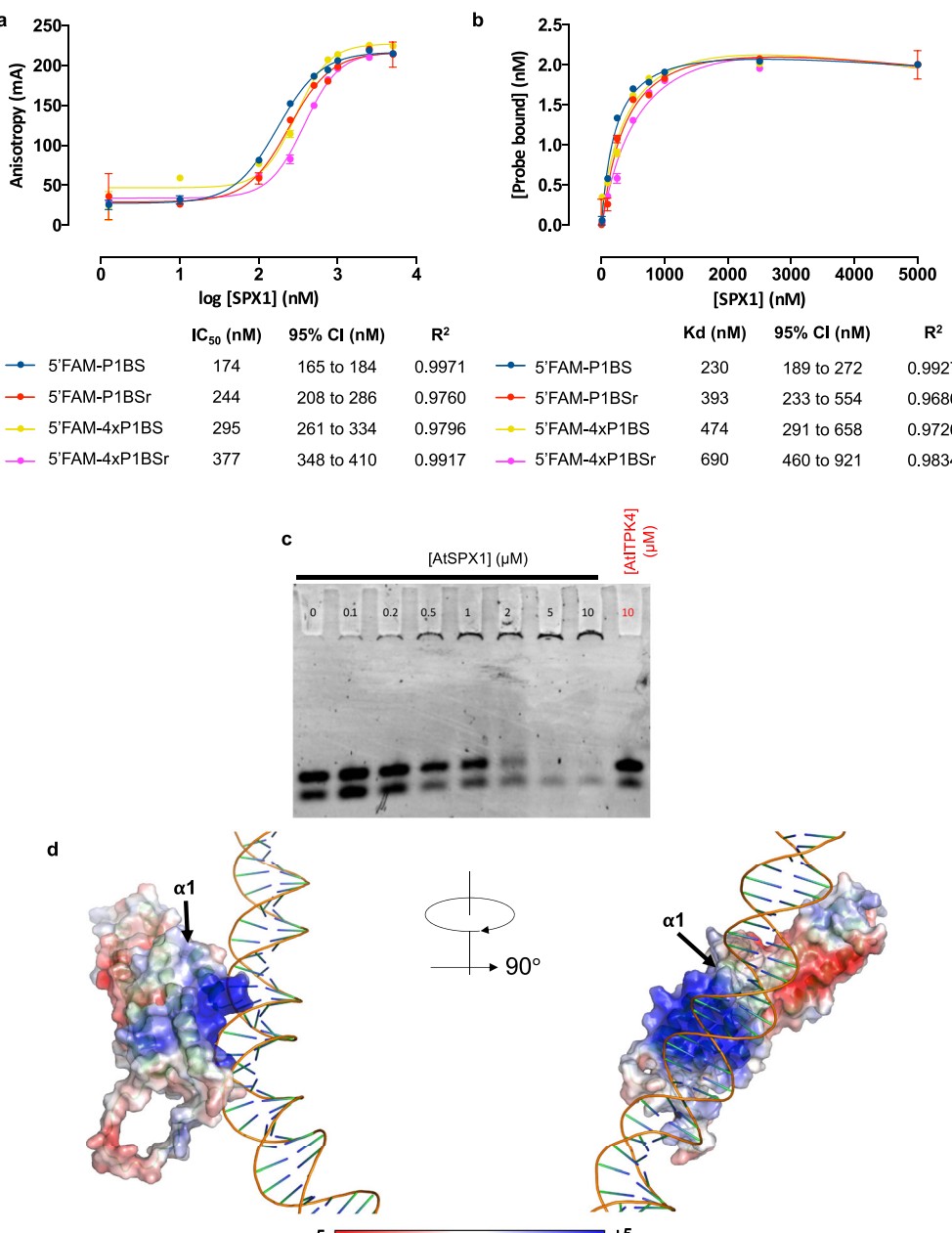

**Fig. 2 | AtSPX1 is a DNA-binding protein. a** AtSPX1 binds a range of fluorescently-tagged dsDNA, indicated by increasing anisotropy as AtSPX1 concentration increases. $N = 4$ replicates, repeated three times with similar results. One-way ANOVA of logIC$_{50}$ revealed a significant difference between all probe pairwise combinations to at least $p < 0.001$, except between 4×P1BS and 4×P1BSr ($p < 0.05$) and between P1BSr and 4×P1BS (ns). **b** Data from a transformed into a fraction of probe bound subjected to a one-site total binding model. **c** EMSA analysis of 5'-FAM labeled ssDNA and dsDNA mix showing 12mer DNA with increasing concentration of full-length AtSPX1. No shift is seen for the control protein AtITPK4 at the highest concentration tested (red). The experiment was repeated twice with similar results. **d** OsSPX1$^{1-198}$ RoseTTAFoldNA model of binding to 4×P1BS (left) and rotated by 90° in the y axis (right), surface colored according to electrostatic charge.

## DNA-bound AtSPX1 is displaced by inositol phosphates and inositol pyrophosphates

Displacement of FAM-P1BS (Supplementary Table 4) from AtSPX1 by InsPs (Fig. 4b) and PP-InsPs (Fig. 4b, c) reveals IC$_{50}$ values approximately double those of 2-FAM-InsP$_5$ displacement (Fig. 1) with a similar pattern of PP-InsP$_5$ and [PP]$_2$-InsP$_4$ molecules displacing the probe more efficiently than InsP$_4$ or InsP$_5$ molecules. Here, because $K_i = IC_{50}/(1+ [ligand]/K_d)^{29}$, IC$_{50}$ approximates very closely to $K_i$. By testing a full, with the exception of 4/6-PP-InsP$_5$, set of potential physiological ligands of full-length SPX1, we observed that, in contrast to the displacement of 2-FAM-InsP$_5$ (Fig. 1), 5-PP-InsP$_4$ was significantly ($p < 0.0001$) less effective in the displacement of FAM-DNA than InsP$_6$, PP-InsP$_5$, or [PP]$_2$-InsP$_4$. Modeling suggested significant repositioning of Y25, K26, and K29 compared to their position when InsP$_6$ is bound. These general trends were repeated when FAM-P1BS was used as the displaced probe. Clearly, individual pyrophosphate substituents make little specific contribution to the strength of ligand binding over and above that of InsP$_6$.

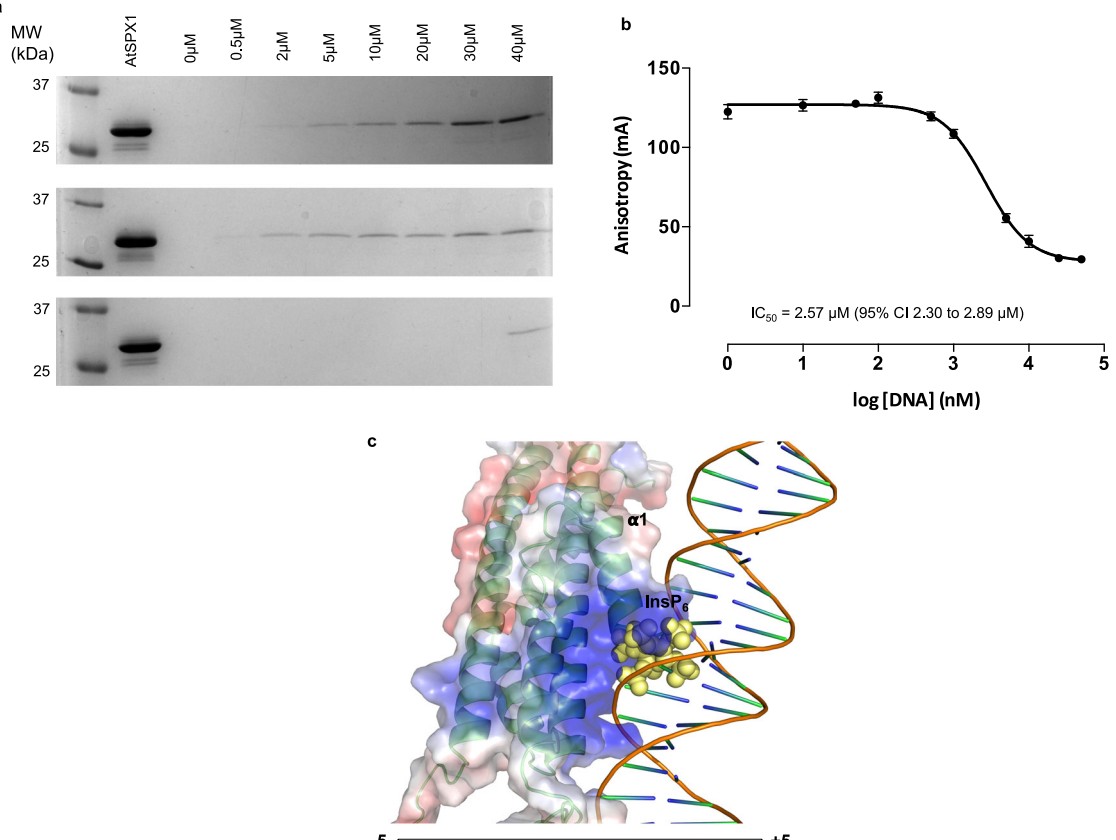

**Fig. 3 | AtSPX1-bound to an inositol phosphate matrix is displaced by DNA.**
**a** Full-length AtSPX1 is displaced from a resin-immobilized InsP$_5$ (linked via the 2-oxygen atom) by successive aliquots of increasing concentration of InsP$_6$ (top) or P1BS DNA (middle), but not by buffer alone (bottom). Repeated twice with similar results. **b** Displacement of 2-FAM-InsP$_5$ from full-length AtSPX1 by increasing DNA concentration. Probe (2 nM) was incubated with 300 nM SPX1 in the presence of varying concentrations of P1BS DNA. $N = 4$ replicates, repeated three times with similar results. **c** Proposed RoseTTAFoldNA[24] model of OsSPX1$^{1-198}$ bound with 4× P1BS DNA showing the overlap in binding position of InsP$_6$ (yellow spheres) and DNA. The protein surface is colored according to electrostatic charge.

## Discussion

In Arabidopsis, phosphate starvation reduces the levels of inositol phosphates and diphosphoinositol phosphates[20], and the roles of both in biotic and abiotic stress responses have been reviewed[30]. While the role of Pi as 'molecular signal' of PSR[6] has been replaced by InsP$_6$ or PP-InsPs[4,8,10,22], not without some debate[28], it should not be forgotten that phosphate starvation reduces multiple organic phosphate species. The special role of PP-InsPs as agents of PSR, confusingly, arises from phosphate-resupply experiments in which PP-InsPs rise in a more pronounced manner than InsPs, notwithstanding that they all rise, as do other organic phosphates. It seems unlikely that recovery from phosphate deprivation is simply a reversal of cellular response to phosphate starvation or that the only active ligand of SPX is the least abundant of inositol phosphate or inositol pyrophosphates.

Against this debate, it is worth examining the properties of SPX proteins and of PHR1. Crystallography has shown how the isolated coil-coil (CC) domain of PHR1 forms a dimer[10]. PHR1 interaction with SPX proteins has been demonstrated in pull-down, gel-filtration, small-angle X-ray scattering (SAXS), ITC, and grating-coupled interferometry experiments[6,8,9,22]. There is, however, no a priori reason why PHR1 should be the only cognate partner of SPX1. Indeed, He et al.[31] have recently described the interaction of OsSPX1/2 with OsBZR1, while SPX4 partners include both PHR1 and Production of anthocyanin pigments 1 (PAP1)[32]. There is, additionally, mixed evidence as to the nature of organic or inorganic phosphate ligands that facilitate the interaction of the AtSPX1 protein with PHR1.

The surface cluster of positive residues that constitute an inositol (pyro)phosphate binding site, first identified in the SPX-domain of the Vtc complex of the filamentous fungus *Chaetomium thermophilum*[4], occupies a much larger surface than needed to bind inositol (pyro) phosphate; with what else might they interact? Restricting ourselves to inositol (pyro)phosphate binding to plant SPX1, recent crystallographic evidence indicates that InsP$_6$ (added into crystallizing solution) binds in a pose that engages a basic surface between helices α1, α2, and α4 as well as residues of the mobile helix α1[8] that are posited[4,22] to be important in conferring PP-InsP-specific function on SPX1. It is worth mentioning that PP-InsPs have not been crystallized with plant SPX proteins.

Close inspection of the available crystallographic data for SPX1 PDB: 7E40 and SPX2 PDB: 7D3Y shows, however, for the former, that contacts are made by InsP$_6$ with a serine residue in the linker in the fusion protein that was crystallized. The requirement of the use of a lysozyme fusion reflects the much-vaunted difficulty of handling SPX1. The authors[8] reported two other InsP$_6$ ligands in the asymmetric unit, one that participates in crystal packing, and another bound in a different pose to a second SPX1 molecule. For this SPX1 molecule, density for the mobile helix α1 and adjunct serine is not modeled. This suggests, perhaps, that the mobile helix α1 is not critical for InsP$_6$ binding, a result confirmed by ITC on deletion of helix α1 (Fig. 4a of ref. 8). Indeed, deletion of the α1 helix of otherwise full-length SPX1 retains the dimeric form of SPX1-PHR2, using two molecules of each protein (analyzed by SEC, Supplementary Fig. 8 of ref. 8). This occurs even in the presence of InsP$_6$, which is a prerequisite (in the authors' model) for repression of PHR2-mediated PSR. Put another way, if for full-length SPX1 the α1 helix is a restraint to the 'preconditioning' SPX1 dimerization that separates SPX1 dimer from PHR2, allowing PHR2

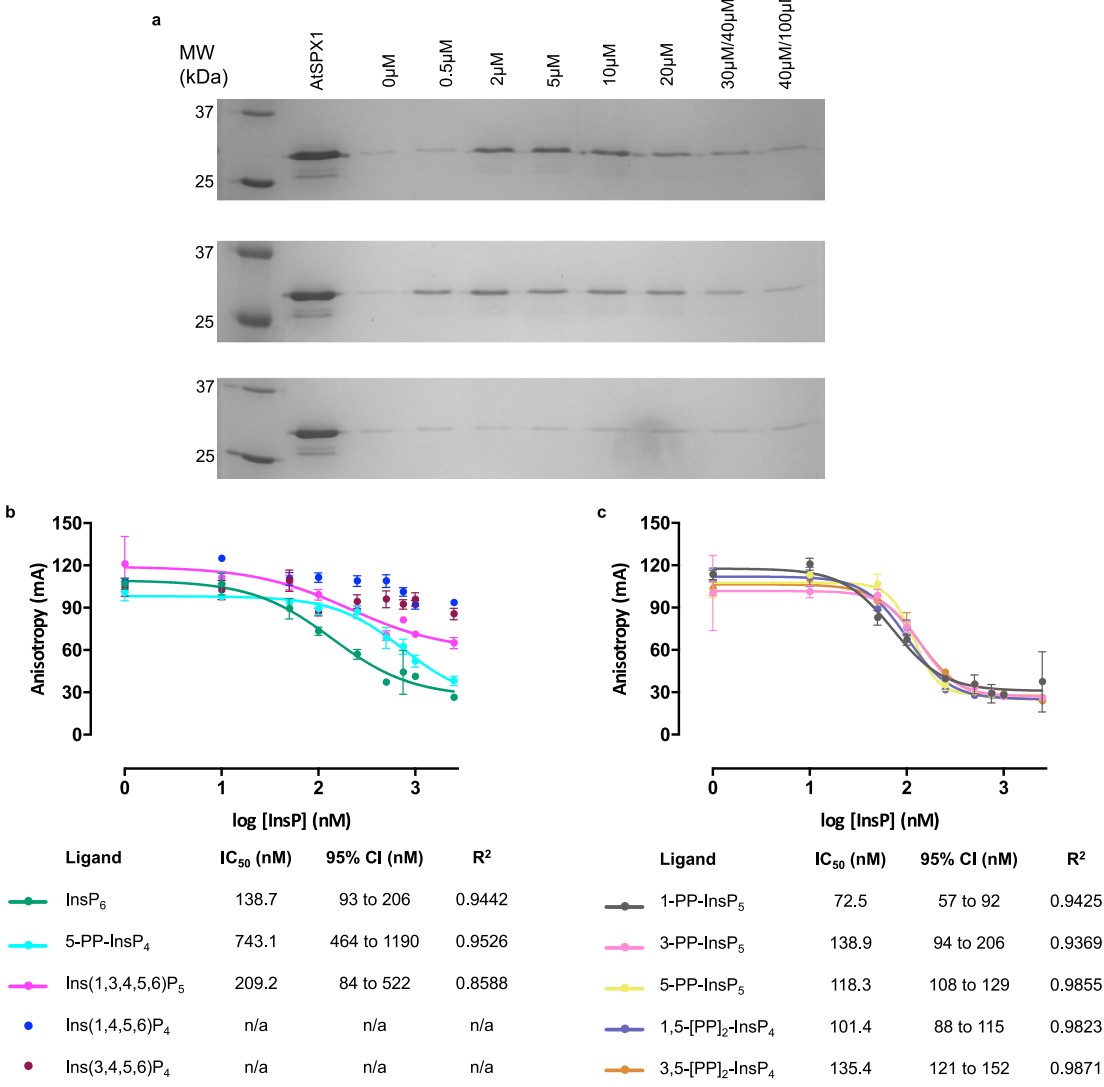

**Fig. 4 | DNA-bound AtSPX1 is displaced by inositol phosphates. a** Full-length AtSPX1 is displaced from a P1BS DNA matrix by successive aliquots of increasing concentration of $InsP_6$ up to 100 μM (top) or P1BS DNA up to 40 μM (middle), but not by buffer alone (bottom). Repeated twice with similar results. **b** Displacement of FAM-P1BS from full-length AtSPX1 by increasing inositol polyphosphate or inositol pyrophosphate concentration. Assay performed in triplicate and repeated twice with the same result. **c** Displacement of FAM-P1BS from full-length AtSPX1 by increasing inositol pyrophosphate concentration. Probe (2 nM) was incubated with 500 nM SPX1 in the presence of varying concentrations of InsP or PP-InsP ligand. One-way ANOVA of $logIC_{50}$ revealed a significant difference between the binding of 5-PP-$InsP_4$ and all other ligands to $p < 0.0001$. Other pairwise comparisons did not show a significant difference. For $InsP_4$ and $InsP_5$, fitting to the 4-parameter logistic was imprecise, reflected in the wide confidence interval for $InsP_5$ and the lack of fit for $InsP_4$ species, $N = 4$ replicates, repeated twice with similar results.

dimer to activate gene expression, then in the absence of $InsP_6$, the factor maintaining SPX1 in dimeric form could be DNA.

Current models of SPX: PHR interaction are reliant on modeling the superposition of the CC domains of separate PHR1 molecules. The separate crystallographic models of SPX: PHR interaction, which do not include AtSPX1 residues 199–259 or OsSPX2 residues 203–280, either do not take account of the encoded lysozyme of the SPX1 fusion protein[8] or involve swapping of the orientation of the monomers for SPX2. For the latter, helices α1 and α2 of the N-terminal domain of one monomer interact with helices α4 and α5 of the C-terminal domain of the other monomer, linked by two antiparallel helices α3 from both monomers[22]. For the former model, superposition of the CC domain of AtPHR1 onto truncated SPX1 places the CC: CC interface (necessary for PHR dimerization) in a position incompatible with the AlphaFold prediction for the missing residues (aa 199–259) of SPX1, while for the latter model, residues 203–280 are not considered. Recently, interrogation of the Vtc complex of yeast, including thermal stability

experiments, has revealed dynamic changes in a previously undescribed α7 helix in the C-terminal region of the SPX domain of Vtc2 subunit[21]. The corresponding region in OsSPX1 is not included in the truncated crystal structure (PDB: 7E40), highlighting the importance of further study of the full-length plant SPX proteins.

The literature offers limited data sets on the relative affinities (as measured by a singular approach) of inositol (pyro)phosphate ligands for manipulable SPX1, while for OsSPX2, no quantitative parameters were reported[22]. We have, therefore, examined in some detail the protein purification requirements demanded by full-length AtSPX1. Here, we reiterate that most studies of plant proteins are of truncated proteins and/or fusions and that the characterization of ligand binding to SPX1 in the absence of other proteins is limited, giving $K_d$ 5–6 μM for $InsP_6$ by ITC[8]. In Pi-deplete scenarios in which SPX1 protein is most strongly expressed, PP-InsPs remain a small fraction of total inositol (pyro)phosphates[20]. Without the partition of InsPs from PP-InsPs, PP-InsPs will be outcompeted by $InsP_6$. Vacuolar compartmentation of

InsP$_6$[33], mediated by MRP5 (ABCC5)[34], provides a mechanism by which competition could be obviated. Alternatively, the close association of ITPK1 function with the activity of SPX1 targets (PHR1) might in mutants[20], be indicative of localized InsP$_6$ and/or PP-InsP production in the vicinity of SPX.

Employing a wider set of inositol (pyro)phosphate ligands, bearing between 4 and 8 phosphates, we find that SPX1 offers no discrimination between 1,5-[PP]$_2$-InsP$_4$, 3,5-[PP]$_2$-InsP$_4$, 1-PP-InsP$_5$ or 3-PP-InsP$_5$, but binds these ligands with slightly lower $K_d$ ($K_i$) than it does InsP$_6$. Indeed, binding affinity is a function of total charge. Protein with a lower A$_{260}$:A$_{280}$ ratio binds the inositol phosphate probe 2-FAM-InsP$_5$ more tightly than protein with the higher A$_{260}$:A$_{280}$ ratio. This led us to speculate that SPX1 binds nucleotide. Such a premise is formally discounted[8] and, for SPX2, is not considered in domain-swap models[22].

Formal testing of the premise that SPX1 binds DNA has been demonstrated by multiple orthogonal approaches: EMSA, non-denaturing PAGE of protein: DNA complexes, fluorescence polarization of FAM-labeled P1BS DNA binding to SPX1, displacement of bound inositol phosphate analog (2-FAM-InsP$_5$) by DNA, affinity purification of SPX1 on resin-immobilized DNA (and displacement by InsP$_6$) and affinity purification of SPX1 on resin-immobilized InsP$_5$ (and displacement by DNA).

The results proffer a fundamentally different perspective of SPX-involvement in PSR (Fig. 5). One in which inositol (pyro)phosphates and DNA are reciprocally competing ligands for binding to SPX. In other words, SPX1-binding to DNA (the less mobile partner) is modulated by (highly mobile) inositol (pyro)phosphates—InsP$_6$ or PP-InsPs. In one possible iteration of this model, depletion of inositol (pyro)phosphates on phosphate starvation increases SPX1 binding to DNA. The extent to which SPX1 binds to inositol (pyro)phosphate or DNA is a function of the binding constants of competing processes and the prevailing concentrations of different inositol (pyro)phosphate ligands. PHR1 binds to the P1BS element of PSI genes, and these include SPX1, but SPX1: DNA interaction is of similar affinity to PHR1: DNA.

The model (Fig. 5) solves the conundrum identified by Collins et al.[35] that SPX1/2 gene transcription is induced by Pi-starvation[6], yet SPX1: PHR1 interaction occurs in Pi-replete conditions (high levels of inositol (pyro)phosphates), i.e., conditions of recovery from PSR. By conferring a specific low Pi function for SPX1, DNA-binding, SPX1 gains molecular function consistent with SPX-dependent phenotypes. The model, moreover, is consistent with evidence that SPX1 is nuclear localized (like SPX2)[5,6,36].

SPX1 and SPX2 show redundancy[6,37,38]. *Spx1/spx2* double mutants are severely growth retarded and possess very low levels of Pi in shoots[38]. In contrast, the *spx4* mutant, another stand-alone SPX protein, accumulates Pi in shoots[37,39]. It seems likely that the precedent of DNA-binding applies to SPX4. In mixture with PAP1, SPX4 was shown by electrophoretic mobility shift assay (EMSA) to bind to the Myb-Recognizing Element (MRE1) of dihydroflavonol-4-reductase (*DFR*) promoters. Binding was also competed by InsP$_6$[39]. Clearly, inositol (pyro)phosphates have broad roles beyond mediation of SPX1-PHR interaction in Pi-replete scenarios[6,38]. Through SPX4, they control expression of PHR1-dependent PSR genes and control PHR1-independent responses[35,37]. Here, although SPX4 is not as well characterized (as SPX1) as an inositol (pyro)phosphate receptor, it is positively responsive to phosphate resupply to Pi-limited plants, it shows dynamic turnover, it acts to retain PHR1 in the cytosol and modulates the shoot PSR[37,39].

Nonetheless, returning to SPX1, the inclusion of missing amino acid residues in our structural model of SPX1: inositol (pyro)phosphate interaction accommodates both InsP$_6$ and PP-InsPs as ligands of the full-length protein. The model challenges the relevance of CC: CC interaction models of dimerization of (truncated) PHR2[8] or SPX2 dimerization (SPX2 PDB: 7D3Y)[22] to full-length SPX proteins.

The model explains how InsP$_6$ and DNA are competing cognate partners of SPX1, sharing the same binding surface on SPX1. The model accommodates differential expression of SPX1 and PHR1 (PHR1 does not respond to phosphate deprivation[36]) and, through differential binding poses of InsP$_6$ and PP-InsP ligands to SPX1, allows for differential interaction of SPX1: PHR1 complexes with potential unidentified partners (as suggested[4]) by modulation of the positive binding surface. While current models do[8] or do not[22] invoke a role of InsP$_6$ as 'molecular glue', both appraise InsP$_6$ as a moderator of PHR: SPX interaction (both describe crystallographic coordination of InsP$_6$). PHR: SPX interaction occurs in the absence of added InsP$_6$[6,9], albeit this interaction is strengthened by added InsP$_6$[8]. For truncated PHR2 $^{230\text{-}426}$, PHR2: SPX1 interaction analyzed by gel-filtration was increased after prolonged incubation of protein with c. 20-fold molar excess (1 mM) of InsP$_6$ (Fig. 1a of ref. 8), against $K_d$ of 5.5 μM for SPX1: InsP$_6$ measured by ITC (Fig. 2b of ref.[8]). In this study, a L348A/ L358A/I362A form of PHR2$^{230\text{-}426}$ (neither full-length nor native protein was reported) bound P1BS DNA with $K_d = 0.26$ μM and bound SPX2 with $K_d \sim 0.17$ μM. Interaction with DNA was largely unchanged ($K_d = 0.38$ μM) by SPX1, but was abolished by additional inclusion of InsP$_6$ at unspecified concentration (Fig. 1e, of ref. 8). The authors posited that PHR2 monomerization induced by InsP$_6$ is a critical step in SPX1- and/or InsP$_6$-induced inactivation of the transcription factor. A similar dependency of SPX: PHR interaction on InsP$_6$ (1 mM) was reported for full-length OsSPX2, co-expressed with truncated OsPHR2 before pull-down. Without knowledge of protein concentration, and hence mol ratio to InsP$_6$, it is difficult to make a comment on the enhancement of the affinity of OsSPX2 for OsPHR2 mediated by InsP$_6$.

While a broader understanding of whether SPX1, or any other SPX protein, has specific 'target' DNA sequences awaits clarification, comparison of $K_d$ InsP$_6$ and $K_d$ DNA with DNA-binding constants for other proteins is informative. A $K_d$ InsP$_6$ of 4.6 μM was reported for the dual-domain protein SPX$_{VTP1}$[40]. We obtained IC$_{50}$ InsP$_6$ (for displacement of 2-FAM-InsP$_5$ from AtSPX1) of 82 nM. For competitive binding, the limits of $K_i$ (effectively $K_d$) are 0.5–1 × IC$_{50}$[29]. In the experiments described here, IC$_{50}$ is approximately equal to $K_i$, because the protein was held at 2 × $K_d$ with ligand 2 orders of magnitude lower. For SPX1, we obtained $K_d$ FAM-P1BS of 235 nM. This value is not far removed from that (88 nM) obtained for the binding of P1BS by PHR1[41]. A value of 16 nM was reported for the binding of the ORE1-NAC transcription factor to its cognate DNA binding site[42]. Qi et al.[7] estimated $K_d$ for binding of MBP-tagged truncated PHR1 (AtdPHR1$^{208\text{-}362}$) to 1×P1BS and 2×P1BS at 707 and 14 nM, respectively. Set against the widely accepted perspective that inositol (pyro)phosphates are cognate ligands of SPX proteins, it is inescapable that the DNA and InsP/PP-InsP binding constants (of SPX1) are similar and of magnitude comparable to the DNA: protein interactions (e.g., of P1BS: PHR1) that support existing mechanistic models of PSR.

The demonstration of DNA- and P1BS-binding by SPX1 raises questions about the topology of interaction of SPX1 and PHR1 with the cognate DNA partner. While an increase in the number of P1BS repeats increased the affinity of PHR1 to DNA, the reduction of $K_{off}$[7] for AtSPX1 reduced the affinity seen on 4×P1BS promoters (Fig. 2). The binding of SPX1 to a single P1BS sequence motif may sterically preclude binding of another SPX molecule to a closely proximal P1BS sequence. Thus, differential expression of PHR1 targets may depend on P1BS repetition. More excitingly perhaps, SPX1 might modulate PHR1/2 function by control of DNA topology during transcription of PSR genes. We draw support from the role of a distal, surface (non-catalytic) DNA binding site of DNA topoisomerase I, comprising a quartet of lysine and additional serine and arginine residues, which is implicated in supercoiling. This provides precedent for protein binding at juxtaposed DNA segments[43], which could for SPX1 (as dimer) represent separated P1BS motifs. Equally, SPX1, through interaction with PHR1, might be considered a 'proximity sensor' for PHR interaction with DNA.

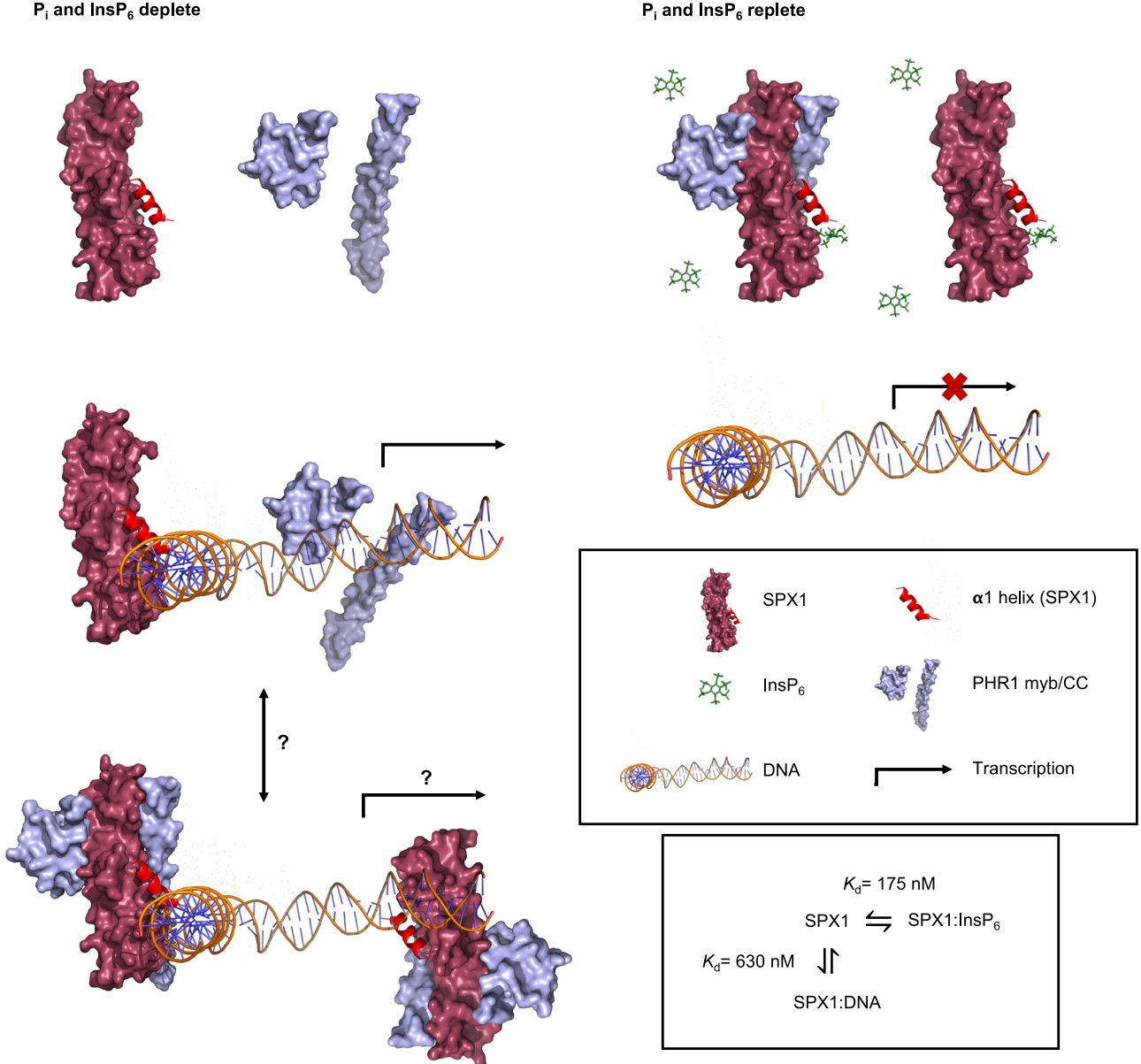

**Fig. 5 | A model of inositol phosphate involvement in PSR that accommodates competition with DNA, and which explains the 'missing' low phosphate role for SPX1.** PSR, the induction of a multitude of Phosphate Starvation Induced (PSI) transcripts, occurs in conditions of low cytosolic phosphate, $Pi_{cyt}$, conditions in which expression of the SPX1 gene, encoding a nuclear-localized protein, is maximal. Inositol phosphates and inositol pyrophosphates are reduced as $Pi_{cyt}$ falls. SPX1 binds DNA; this may include P1BS elements of promoters of PSI genes, including SPX1, forming a scaffold for interaction with PHR1, a MYB-CC transcription factor. The extent to which PHR1: SPX1 protein interaction occurs in the absence or at low levels of inositol (pyro)phosphates is not well-defined, but PHR1, in association or not with SPX1, activates P1BS-driven PSI gene expression. Resupply of phosphate increases $Pi_{cyt}$ with ensuing increase in inositol phosphates and inositol pyrophosphates. $InsP_6$ displaces SPX1 from DNA, including P1BS promoter elements, the mixed equilibria of PHR1: SPX1, PHR1: P1BS and SPX1: DNA interactions are shifted towards SPX1: $InsP_6$ and SPX1: $InsP_6$: PHR1. Displacement of SPX1 from DNA, including P1BS elements, and PHR1 from P1BS ceases the transcription of PSI genes. Model images of SPX1 and PHR1 were generated from (PDB: 7E4O)[8]. We draw support for our model from observations that deletion of the α1 helix facilitates an SPX dimer PHR1 dimer interaction that is not disrupted by $InsP_6$[8]. Perhaps, in the absence of $InsP_6$, SPX1 dimer PHR1 dimer interaction, that of full-length SPX1, is stabilized by binding of SPX1 to DNA.

In Fig. 6, we summarize how exchange of inositol phosphates between subcellular compartments adds organellar dimension to nuclear, vacuolar and cytosolic processes regulated by inositol (pyro) phosphates, as impacted by the newfound DNA-binding function of the SPX1 transcriptional repressor. To date, the only verified inositol phosphate transporter is MRP5[34]. Disruption of Mrp5 may isolate $[PP]_2$-$InsP_4$ and PP-$InsP_5$ species from competition by $InsP_6$, but this needs further elaboration. Both *mrp5* and wild-type (Col-0) show common elevations of $[PP]_2$-$InsP_4$ in shoots on phosphate resupply following starvation, to levels that are a consistent and small fraction of $InsP_6$[20].

Moreover, the two genotypes exhibit similar levels of Pi in shoots, irrespective of Pi supply[20]. MRP5 aside, the dynamic Pi-status-dependent turnover of SPX4[37] makes it likely that SPX4-dependent retention of PHR1 in the cytosol of shoots is itself an inositol (pyro) phosphate function in Pi-replete scenarios.

Finally, others[30,35] have reviewed the engagement of SPX proteins in the interplay of phosphate starvation and growth factor signaling, including COP signalosome function, in plants. Our findings offer perspectives on inositol (pyro)phosphate involvement beyond the role of inositol (pyro)phosphates as 'molecular glue'. Indeed, taking jasmonate

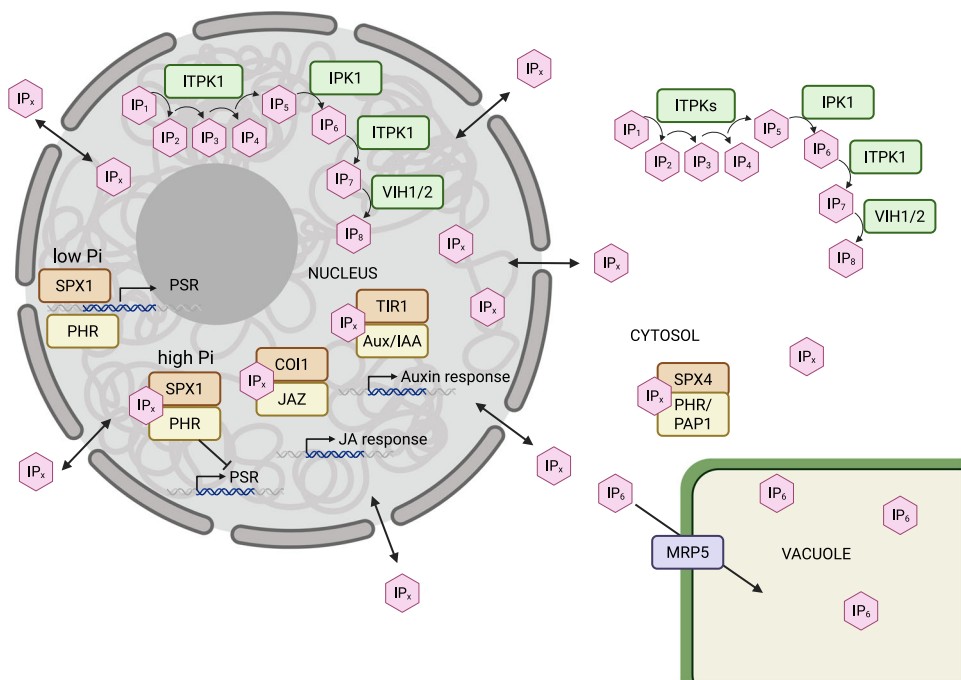

**Fig. 6 | A simplified schematic of InsP metabolism and transport between organelles (nucleus, vacuole) and cytosol in a plant cell.** Highlighting IPKs that have been shown to localize in the nucleus from the lipid-independent pathway of InsP biosynthesis (green) and InsP binding proteins (orange) with protein partners (yellow). InsPs (IPs) are shown as pink hexagons. Straight arrows represent exchange of InsPs between subcellular compartments, curly arrows represent phosphorylation of InsPs. Created in BioRender. Brearley, C. (2025) https://BioRender.com/1z942ml.

as an example, for cells with low levels of jasmonate and its bioactive conjugate JA-Ile, Jasmonate–ZIM–Domain (JAZ) proteins repress the transcription factors that regulate JA-responsive genes. JA-Ile promotes binding of JAZ proteins to the jasmonate receptor CORONATINE INSENSITIVE1 (COI1), which is an F-box protein that, as part of the SCF[COI] E[3] ubiquitin ligase complex, targets proteasomal degradation of JAZ repressors[44]. The inositol (pyro)phosphate dependence of the interaction of JAZ proteins with COI1 (reviewed[30]) critically places jasmonate signaling responsive to phosphate status[35]. The finding of competition between inositol (pyro)phosphates and DNA (promoter elements) likely impacts the network of signaling interactions of biotic and abiotic stress pathways. These hierarchies will include transcriptional phenomena and the protein: protein interactions that underlie them. It is unlikely that direct competition between DNA and inositol (pyro)phosphates is restricted only to PSR. Our methods, particularly fluorescence polarization and affinity matrices, offer an opportunity here.

## Methods
### Protein expression and purification
Arabidopsis thaliana SPX1 (At5g20150) was cloned into the pOPINF vector using primers 5′-AAGTTCTGTTTCAGGGCCCGATGAAGTTT GGTAAGAGTCTCA-3′ and 5′-ATGGTCTAGAAAGCTTTATTTGGCTTCT TGCTCCAA-3′. The In Fusion HD enzyme kit (Clontech) was used to recombine HindIII and KpnI-digested pOPINF plasmid with the PCR product. This was transformed into *Escherichia coli* Stellar cells (Clontech) and colonies confirmed by PCR amplification. The construct was transformed into ArcticExpress™ RIL cells, with expression cultures grown at 13 °C, 0.3 mM IPTG for 24 h. Sequence provided in source data.

Lysate recovered from a French press of the cell pellet in 50 mM NaH$_2$PO$_4$, pH 6.5, 300 mM NaCl, 20 mM imidazole, 0.5% Triton X-100, was applied to a 5 mL nickel nitriloacetic acid (Ni−NTA) HiLoad column (Qiagen) equilibrated in 50 mM NaH$_2$PO$_4$, pH 6.5, 300 mM NaCl, 20 mM imidazole. The column was washed with 50 mM NaH$_2$PO$_4$, pH 6.5, 1 M NaCl, 20 mM imidazole before applying a gradient of Ni−NTA

buffer A (50 mM NaH$_2$PO$_4$, pH 6.5, 300 mM NaCl, 20 mM imidazole) to Ni-NTA buffer B (50 mM NaH$_2$PO$_4$, pH 7.5, 300 mM NaCl, 500 mM imidazole). Subsequently, the 6× His tag was cleaved using HRC 3 C protease (ThermoFisher Scientific Pierce) at 4 °C overnight in buffer A. Pooled fractions were diluted with 20 mM Tris-HCl pH 7.5 and applied to a 1 mL Heparin HiTrap column (Qiagen) equilibrated with heparin A buffer (20 mM Tris-HCl pH 6.5, 50 mM NaCl), washed with heparin A buffer, and eluted with a gradient of heparin A to heparin B buffer (20 mM Tris-HCl pH 6.5, 1 M NaCl). Protein was buffer exchanged into 10 mM sodium phosphate, pH 6.5, 600 mM NaCl, 6 mM DTT, 10% glycerol for storage at −80 °C. Protein concentration was quantified by absorbance at 280 nm using the NanoDrop One (Thermofisher) and calculated using the extinction coefficient generated by Protparam (Expasy). The level of nucleotide contamination was observed using the Nanodrop One (Thermofisher) A$_{260: 280}$ ratio. Protein was purified on four separate occasions; the initial purification was split into two, either without (prep A) or with (prep B) the high salt wash step after protein loading. The three subsequent purifications were performed as described above. Expression and purification of AtITPK4 (At2g43980) and HsSHIP2 (Uniprot: O15357) for use as control proteins were performed as described previously[14,45].

### Fluorescence anisotropy and probe synthesis
Assays were performed not less than three times as described[46] in 20 mM 4-(2-hydroxyethyl)-1-piperazineethane sulfonic acid (HEPES), pH 6.5, 1 mM MgCl$_2$, 100 mM NaCl. Probes were used at 2 nM in 4 replicates of 20 µL volume in Corning nonbinding 384-well plates (product no. 3575). Fluorescence polarization was recorded on a BMG ClarioSTAR plate reader: excitation 485 nm, 12 nm; dichroic 505 nm; emission 505 nm, 16 nm; and 200 flashes. Individual run settings in the source data. For displacement assays, protein was used at ≈ 2 × $K_d$, 300 nM with 2-FAM-InsP$_5$ or 500 nM with 5′-FAM-DNA. All assays were performed at room temperature. IC$_{50}$ was determined using a 4-parameter fit of anisotropy data and statistical significance using a one-way ANOVA on logIC$_{50}$ values, using GraphPad software.

2-FAM-InsP$_5$ was synthesized from 2-$O$-(2-aminoethyl)-Ins(1,3,4,5,6) P$_5$ as previously reported[47]. 5-FAM-InsP$_5$ was synthesized in a similar way from *myo*-inositol 5-(3-aminopropylphosphate) 1,3,4,6-tetrakisphosphate (Supplementary Fig. 1). Briefly, 5-FAM-InsP$_5$ consists of Ins(1,3,4,5,6)P$_5$ coupled to 5-carboxyfluorescein via a linker attached to a terminal oxygen atom of the 5-phosphate group of the InsP$_5$. This isomer of InsP$_5$ is a *meso*-compound, with a plane of symmetry through C-2 and C-5 atoms and the synthesis of conjugates is therefore simplified by functionalising the InsP$_5$ at either $O$-2 for 2-FAM-InsP$_5$ or $O$-5 for 5-FAM-InsP$_5$; see Supplementary Methods Fig. 5 for full details.

### Protein modeling
Induced Fit docking (IFD), followed by molecular dynamics (MD) simulations, was performed on each ligand, and MMPBSA calculations were used to estimate binding free energy. Computational alanine scanning mutagenesis was also performed. Methods for the process can be found in Supplementary Methods.

RoseTTAFold2NA[24] was used to predict the structures of protein: nucleic acid complexes. Default parameters were adopted to predict structures based on the full-length amino acid sequences of Arabidopsis (Uniprot: Q8LBH4) and rice (Uniprot: Q69XJ0) SPX1 or the truncated SPX$^{1-198}$ sequences of both, with the 54 bp double-stranded nucleic acid sequence of P1BS repeat (4×P1BS) (Supplementary Table 4).

### Electrophoretic mobility shift assay
Binding assays were performed in 20 µL reactions with 2.5 nM FAM-oligomer and SPX1 (0–10 µM) or AtITPK4 (10 µM) as a negative control, in 20 mM HEPES pH 6.5, 1 mM MgCl$_2$, 50 mM NaCl (or 100 mM where stated), 0.1 mg/mL BSA, 1 mM ethylenediamine tetra-acetic acid (EDTA). Samples were incubated on ice for 1 h before adding native loading dye (50 mM Tris-HCl, pH 6.8, 0.01% (w/v) bromophenol blue, 10% (v/v) glycerol). An 8% native polyacrylamide gel (8% acrylamide/bisacrylamide, 375 mM Tris pH 8.8, 0.1% APS, 0.05% TEMED) with 5% native stacking gel (5% acrylamide/bisacrylamide, 125 mM Tris pH 6.8, 0.1% APS, 0.05% TEMED) was pre-run at 50 V for 2 h on ice. Samples were loaded and run at 150 V for a further 2–2.5 h on ice. Gels were imaged using a Typhoon™ FLA 9500 system (GE Healthcare) using the preset parameters for FAM detection. Experiment performed three times with varying dsDNA probe sequence and SPX1 concentrations, binding consistent using clean AtSPX1 stocks.

### Pull-down assays and synthesis of 2-linked InsP$_5$ affinity matrix
The 2-linked InsP$_5$ affinity matrix was produced by reaction of 2-$O$-(2-aminoethyl)-Ins(1,3,4,5,6)P$_5$[47] with Affi-Gel 10 (Biorad, UK) using the non-aqueous coupling methodology previously described for 2-$O$-(2-aminoethyl)-Ins(1,4,5)P$_3$[48]. Trials showed that this method typically gave ≥2 µmole of immobilized InsP$_5$ per mL of settled gel as determined by total phosphate assay, while much lower coupling efficiencies were obtained when aqueous buffers were used. Full details are given in Supplementary Methods.

Three aliquots of 2-linked InsP$_5$ beads were washed three times with assay buffer (20 mM HEPES, pH 6.5, 50 mM NaCl), 7 µM AtSPX1 was added to each and incubated on ice with gentle agitation for 1 h. An increasing concentration of InsP$_6$ or dsDNA was added (or the same volume of buffer as a control) and incubated on ice for 15 min after each addition, with a sample of the same volume as the addition taken for analysis before the next addition was made. Samples were analyzed by SDS-PAGE using 12% acrylamide gels at 150 V for 1 h and stained using InstantBlue® Coomassie stain (Abcam). The experiment was performed twice with varying concentrations of InsP$_6$ or dsDNA.

### Pull-down assays using P1BS-bound matrix
P1BS-bound Sepharose beads were generated using high-capacity Streptavidin Sepharose™ High Performance beads (Cytiva 17-5113-01, obtained from Merck, UK). Beads were washed three times with binding buffer (20 mM Tris, pH 7.5, 150 mM NaCl, 1 mM EDTA) and resuspended with an additional 200 µL binding buffer containing 20 µM biotin-tagged P1BS dsDNA (MWG Eurofins). Beads were incubated for 1 h at room temperature with gentle agitation. A separate sample of beads was incubated with buffer only, to be used as a control. After 3 washes with assay buffer (20 mM HEPES pH 6.5, 50 mM NaCl), 7 µM AtSPX1 was added and incubated on ice with gentle agitation for 1 h. Initially, protein-bound DNA-tagged (DNA-conjugated) beads and control beads were analyzed by boiling a small sample of beads in SDS loading buffer, with subsequent addition to a final concentration of 100 µM InsP$_6$ or an addition of the same volume of water as a control and incubation on ice for 30 min. Both supernatant and bead samples were analyzed by SDS-PAGE. Alternatively, DNA-conjugated or control beads were split into aliquots and either an increasing concentration of InsP$_6$ or dsDNA was added (or the same volume of buffer as a control) and incubated on ice for 15 min after each addition, with a sample of the same volume as the addition taken for analysis before the next addition was made. Samples were analyzed by SDS-PAGE using 12% acrylamide gels at 150 V for 1 h and stained using InstantBlue® Coomassie stain (Abcam). Experiment performed twice at varying concentrations of InsP$_6$ or dsDNA.

### Ion exchange HPLC
HPLC quantification of orthophosphate was performed by chromatography on a 2 × 250 mm Dionex™ IonPac™ AS18 column on a Dionex™ ICS-6000 Ion Conductivity System. The column was eluted at a flow rate of 0.5 mL min$^{-1}$ with a gradient of potassium hydroxide delivered according to the following schedule: time (min), concentration (mM); 0, 0; 12, 12; 20, 34; 25, 34; 26, 0; 36, 0. Orthophosphate was detected at retention time 18 min by suppressed conductivity measurement.

### Reporting summary
Further information on research design is available in the Nature Portfolio Reporting Summary linked to this article.

## Data availability
The Molecular dynamics data generated in this study have been deposited in the MDRepo database under accession codes MDR00004441, MDR00004443, MDR00004444, MDR00004445, MDR00004446, MDR00004447, MDR00004448. The free energy calculation data are available at the University of East Anglia Digital Respository. Source data are provided with this paper.

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

## Acknowledgements

We thank Hui-Fen Kuo and Tzyy-Jen Chiou, Academia Sinica, Taiwan, for the gift of plasmid encoding SPX. Induced fit docking calculations and SPX1: DNA complex structure predictions were carried out on the High-Performance Computing Cluster supported by the Research and Specialist Computing Support service at the University of East Anglia. The molecular dynamics simulations and MMPBSA calculations needed to complete the second round review corrections were performed by A.M.H. using the high-performance computing facility at the International Center for Food and Health, Shanghai Ocean University. This research was funded in whole, or in part, by the Natural Environment Research Council (Grant number NE/W000350/1) to C.A.B. and the Wellcome Trust [Grant number 101010] to B.V.L.P. as a Wellcome Trust Senior Investigator. For the purpose of open access, the authors have applied a CC BY public copyright licence to any Author Accepted Manuscript version arising from this submission.

## Author contributions

C.A.B.: conceptualization, funding acquisition, project administration, formal analysis, supervision, investigation, methodology, writing—original draft, writing—review and editing. A.M.H.: investigation, methodology, writing—review and editing. M.G.: investigation. B.V.L.P.: funding acquisition, supervision, methodology, writing—review and editing. A.M.R.: investigation, supervision, methodology, writing—review and editing. M.L.S.: investigation, methodology, writing—review and editing. H.L.W.: formal analysis, investigation, methodology, writing—original draft, writing—review and editing.

## Competing interests

The authors declare no competing interests.
