## [Transparent Peer Review file · Nature Communications]

The intracellular inositol (pyro)phosphate receptor AtSPX1 reciprocally binds to P1BS DNA

Corresponding Author: Professor Charles Brearley

Version 0:

Reviewer comments:

Reviewer #1

(Remarks to the Author)

Whitfield et al. are experts in crystallography and enzymology with focus on enzymes of inositol polyphosphate synthesis. In this manuscript, they apply their knowledge and extensive tool set to the characterization of interactions between Arabidopsis SPX1 and inositol phosphates. Determination of K_D / IC_{50} values for binding of SPX1 - and previously characterized ITPK4 control - to the entire range of InsP4 to InsP8 species found in plants shows increasing affinity with overall negative charge - corroborating earlier findings but with defined molar ratios of protein and ligand - as well as using untagged full-length SPX1 protein. By assessing two different batches of purified recombinant SPX1 protein, the authors for the first time present evidence for direct interaction between SPX1 and DNA - with some - albeit small - preference for P1BS cis elements known for attracting PSR master regulator and MYB-CC transcription factor PHR1 to PSR gene promoters. This presents a significant new finding that may help to explain why SPX1/2 that were initially only thought to suppress PHR1 binding to PSR gene promoters under P-replete condition are among the earliest PSR genes known to date. In Figure 5, the authors present a refined working model(s) for SPX1/2 - PHR1 - DNA interactions with changing cellular inositol phosphate pools. Their findings will help to interpret a number of puzzling findings that the 'PSR community' has been struggling to interpret.

Minor concerns are:

- 1) The over-emphasis on InsP6 as the most likely '*in vivo*' ligand (abstract, line 380 ff., line 407 ff.). It might be worth pointing out that InsP6 (phytate) is largely stored in the vacuole. Riemer *et al.* (2021) propose that InsP8 could be generated more locally by close association of ITPK1 with individual SPX-domain proteins and thus have the ability to quickly convey changes in P status. Kuo *et al.* 2018 (doi:10.1111/tpj.13974) reported a shoot-specific increase in InsP7 following Pi withdrawal that also highlights local differences in InsP pools. From the IC_{50} values shown in Figure 1, one appreciates higher affinity towards higher order polyphosphates.
- 2) Line 185 ff. - please expand on what 'confirms the relevance of this new finding' - this is a bit unclear.
- 3) Line 194 ff. - please add a stronger discussion point to further comment on this result of 'slightly weaker binding' of a randomized sequence compared with P1BS as well as the contrasting behavior of PHR1 towards P1BS? It would be good to pick this up when explaining the model shown in Figure 5.
- 4) Line 269 heading - could you use 'or free P1BS / GC-rich DNA probe' instead of 'or DNA' as this is a bit unclear?
- 5) Line 324 - could add other examples of alternative SPX targets, such as SPX4-PAP1 (<https://doi.org/10.1111/nph.17139>)?
- 6) Line 367 ff., check spelling of vtc / vtc2 - all capital letters (?)
- 7) Line 418 ff., Arabidopsis / rice *spx1 spx2* double mutants show severe growth retardation and very low shoot Pi levels; they do not show 'accumulation of Pi in shoots'. The latter is seen in Arabidopsis *spx4* mutants.
- 8) Line 436, check spelling 'a ^{3M} mutation'
- 9) Figure 5 - move sentence beginning in line 467 to discussion - and expand on this point a little more. This aspect does not appear to be visualized.
- 10) Throughout Materials & Methods section, state the number of replicates for assays such as protein purification, EMSA, pull-downs. Add how protein was quantified to the protein purification section.
- 11) Reference AGI codes for the proteins used in this study.

Reviewer #2

(Remarks to the Author)

The manuscript NCOMMS-25-06431 investigated the binding of InsP, PP-InsP, and DNA ligands to Arabidopsis thaliana SPX1 (AtSPX1) through organic synthesis, molecular interaction experiments and molecular docking. The findings proffer a fundamentally different perspective of SPX involvement in PSR. The current manuscript only investigated the interactions of InsP, PP-InsP with DNA and AtSPX1 through molecular docking. Further experiments are needed, such as molecular dynamics simulation and cryo-electron microscopy. So I think the article should be major revised before publication.

1. First, there is a lack of mass spectroscopic evidence in the PP-InsP organic synthesis section.
2. The focus of this study is on the binding of full-length and truncated SPX1 to PHR or InsP, PP-InsP. It is insufficient to analyze the interaction of the complex of SPX1 with PHR or InsP, PP-InsP using only static molecular docking methods. Structural analysis, or molecular dynamics simulation of the complex is required. Please modify as required.
3. The full-length proteins was used in protein purification and molecular interaction experiments, while the crystallographic structures of truncated SPX1 was employed in molecular docking of SPX1 with PHR or InsP, PP-InsP molecules. In the study, the author mainly studies the differences between SPX1 lacking certain amino acids and full-length SPX1, which seems unreasonable and unconvincing.
4. Regarding the molecular docking results of the interactions between SPX1 and InsP and PP-InsP, the authors did not perform bonding analysis on the key amino acids when visualizing the molecular interactions in the article, as shown in Figure S4. The binding sites of different InsP and PP-InsP with SPX1 could not be exactly the same. The binding sites were quantified by combining bonding analysis with non-bonding interactions and by analyzing the energy contribution value of each amino acid through MMPBSA. Please modify as required.
5. Regarding the determination of affinity, the author has not provided the on-machine or fitted spectra but merely presented the numerical values of affinity. Meanwhile, the method for ITC determination was not provided. Please modify as required.
6. The article lacks statistical explanations.
7. In addition, more information about the significance of this study should be supplemented in the discussion section of the article.

Version 1:

Reviewer comments:

Reviewer #1

(Remarks to the Author)

The authors have addressed all of the previously raised concerns to my satisfaction. I am not a biochemist, so will rely on the second reviewer for his evaluation of the authors' response to his comments. I would say, however, that I do not think that cryo electron microscopy is required to boost the manuscript's impact. The biochemistry approach taken here is a valid characterisation of the system in its own right. Statistical information and the number of replicates has now been included. The modified Figure 6 adds a lot of value to the discussion of the findings.

Just one minor concern:

In Figure 6, please remove the SPX1-PHR1 module in the cytosol. While it has been shown that SPX4-PHR1 interact in the cytosol - and can also shuttle to the nucleus (either together or individually), SPX1 (SPX2)-PHR1 interaction has so far only been observed in the nucleus.

Reviewer #2

(Remarks to the Author)

The manuscript NCOMMS-25-06431 investigated the binding of InsP, PP-InsP, and DNA ligands to Arabidopsis thaliana SPX1 (AtSPX1) through organic synthesis, molecular interaction experiments and molecular docking. Major revisions are still needed before publication.

1. First of all, the introduction section does not clearly explain the reasons for choosing the SPX1 protein or the previous shortcomings. The significance of this study should be emphasized in the introduction section. And the innovative aspects of this article need to be more prominently highlighted.
2. The supplementary kinetic simulation in the article only simulated for 10 nanoseconds, which is completely unconvincing. It should have simulated for more than 100 ns.
3. The kinetic simulation merely analyzed the results of MMPBSA. RMSD, RMSF, SASA, hydrogen bonds and distance evolution are important criteria for evaluating the reliability of the simulation. Please provide a diagram to supplement.
4. The author points out that Inositol (pyro)phosphates and DNA are reciprocally competing ligands of SPX1. The molecular dynamics simulation did not simulate DNA and SPX1. It is necessary to provide evidence to prove this phenomenon.
5. The method for virtual scanning of mutations needs to be supplemented. The header in Table S2 states that it presents the results of scanning mutations, but it also says, "Gbind values, calculated over 100 frames (5 ns) of the molecular-dynamics trajectories, are reported with standard deviation (in units of kJ/mol)." Could you please clarify whether this is the result of scanning mutations or the result of MMPBSA energy decomposition? If it is the result of scanning mutations, could you explain how the deviation values were obtained?
6. The visualizations in Figures 1c and S4 of the article do not specify which specific conformation the simulation was carried out at. Therefore, a FEL analysis should be conducted on the simulation trajectories to identify the most stable

conformation for analysis. The most important point is that based on the visualized graph, neither amino acids nor small molecules' hydrogen were detected. It is very likely that before the simulation, the hydrogen addition treatment for small molecules and proteins was not carried out. Therefore, the results obtained in this way are incorrect.

7. Figure S4 should have its trajectory analysis improved. Analyze the interaction graphs for different time periods, such as 0 ns, 25 ns, 50 ns, 75 ns, and 100 ns.

8. The ITC results in the article only reproduced 8 data points, and no obvious S-shaped pattern was observed as shown in Figure 2B. The results are unreliable. Additionally, please provide detailed measurement methods. At the same time, please attach the original image of the machine operation as an attachment.

9. Figures 5 and 6 are both schematic diagrams. It seems unnecessary.

10. Many of the icons in the article do not have sufficient clarity and their quality has not reached the publication standards.

11. First, there is a lack of mass spectroscopic evidence in the PP-InsP organic synthesis section.

Version 2:

Reviewer comments:

Reviewer #2

(Remarks to the Author)

The author has answered the questions I raised earlier, and I am satisfied with the response.

10 Sept 2025

Dear Reviewers

We thank you for your considered criticisms and suggestions for improvement.

Here we submit a revised manuscript.

To address Reviewer #1 (Remarks to the Author):

Whitfield et al. are experts in crystallography and enzymology with focus on enzymes of inositol polyphosphate synthesis. In this manuscript, they apply their knowledge and extensive tool set to the characterization of interactions between Arabidopsis SPX1 and inositol phosphates. Determination of K_D / IC_{50} values for binding of SPX1 - and previously characterized ITPK4 control - to the entire range of InsP4 to InsP8 species found in plants shows increasing affinity with overall negative charge - corroborating earlier findings but with defined molar ratios of protein and ligand - as well as using untagged full-length SPX1 protein. By assessing two different batches of purified recombinant SPX1 protein, the authors for the first time present evidence for direct interaction between SPX1 and DNA - with some - albeit small - preference for P1BS cis elements known for attracting PSR master regulator and MYB-CC transcription factor PHR1 to PSR gene promoters. This presents a significant new finding that may help to explain why SPX1/2 that were initially only thought to suppress PHR1 binding to PSR gene promoters under P-replete condition are among the earliest PSR genes known to date. In Figure 5, the authors present a refined working model(s) for SPX1/2 - PHR1 - DNA interactions with changing cellular inositol phosphate pools. Their findings will help to interpret a number of puzzling findings that the 'PSR community' has been struggling to interpret.

We are pleased that the reviewer considers our work to represent 'significant new finding' and that the manuscript 'will help to interpret a number of puzzling findings that the 'PSR community' has been struggling to interpret'.

Minor concerns are:

1) The over-emphasis on InsP6 as the most likely '*in vivo*' ligand (abstract, line 380 ff., line 407 ff.). It might be worth pointing out that InsP6 (phytate) is largely stored in the vacuole. Riemer *et al.* (2021) propose that InsP8 could be generated more locally by close association of ITPK1 with individual SPX-domain proteins and thus have the ability to quickly convey changes in P status. Kuo *et al.* 2018 (doi:10.1111/tpj.13974) reported a shoot-specific increase in InsP7 following Pi withdrawal that also highlights local differences in InsP pools. From the IC_{50} values shown in Figure 1, one appreciates higher affinity towards higher order polyphosphates.

We have tempered our enthusiasm for the suggestion that InsP₆ is more likely than InsP₈ to be a/the physiological ligand, by citing evidence that InsP₆ might be compartmentalized from PP-InsPs (see below). This is a neutral position on the reviewer's quoted Riemer et al speculation of close association of SPX proteins and ITPK1.

Moreover, to further emphasize a neutral tone, to allow the data to 'talk', we have tried to use the term 'inositol (pyro)phosphates' throughout, to cover InsP₆ and PP-InsP species alike, except where explicit terminology is demanded.

Nevertheless, the field needs an explicit demonstration that PP-InsPs exert their influence uncrowded by InsP₆. We, therefore, cite Nagy et al (Ref. 33, lines 363 and 472) as the seminal mechanistic discovery of InsP₆ transport, and Mitsuhashi et al (cited as Ref. 32, line 363) as the seminal measurement of InsP₆ in vacuoles (see also, Point 5, below).

We also have undertaken statistical test of the differences (or otherwise) of experimentally determined IC₅₀ values for different ligands (Legend to Figure 1) reported also lines 92 ff.

2) Line 185 ff. - please expand on what 'confirms the relevance of this new finding' - this is a bit unclear.

We apologize for the cryptic nature of this comment. The modified sentence (line 193 ff.) elaborates by explaining how weaker binding affinities reported in the literature are the mechanistic basis of claimed InsP₈ function and how DNA binding of the affinity described should therefore be considered equally mechanistic.

3) Line 194 ff. - please add a stronger discussion point to further comment on this result of 'slightly weaker binding' of a randomized sequence compared with P1BS as well as the contrasting behavior of PHR1 towards P1BS? It would be good to pick this up when explaining the model shown in Figure 5.

We thank the reviewer for this suggestion. We have (in the legend to Figure 2 and lines 200 ff.) provided statistical test of the differences in binding affinities of different DNA probes and have provided the stronger discussion point requested in the Discussion (lines 444 ff., and 454 ff.) in context of the model shown in Figure 5 (as suggested). Statistical test was performed on logIC₅₀ obtained from the 4-parameter logistic fit to the raw polarization data.

4) Line 269 heading - could you use 'or free P1BS / GC-rich DNA probe' instead of 'or DNA' as this is a bit unclear?

We thank the reviewer for this suggestion. We have modified the heading accordingly (line 259), '*P1BS-immobilized AtSPX1 is displaced by inositol phosphates or free P1BS / GC-rich DNA probe*'

5) Line 324 - could add other examples of alternative SPX targets, such as SPX4-PAP1 (<https://doi.org/10.1111/nph.17139>)?

We thank the reviewer for this suggestion, we have modified the sentence (lines 306 ff.) to include SPX4, which, while not well-described as inositol (pyro)phosphate binder, has well-described interaction with protein partners such as PHR1 and PAP1. We cite (He et al. 2021, Ref 31) and return to this point in the Discussion in lines 400 ff., citing review (Collins et al. 2024, Ref 34) and in lines 476 ff., citing primary literature (including Osorio et al 2019, Ref 36).

To better address the stand-alone SPX family, placing discussion in context with inter-organelle communication and InsP₆ transport as factors in PSR, we provide a new figure (Figure 6).

6) Line 367 ff., check spelling of vtc / vtc2 - all capital letters (?)

This has been amended (lines 348 ff., and throughout) to Vtc (as used Pipercevic et al., 2023 N Comms).

7) Line 418 ff., Arabidopsis / rice *spx1 spx2* double mutants show severe growth retardation and very low shoot Pi levels; they do not show 'accumulation of Pi in shoots'. The latter is seen in Arabidopsis *spx4* mutants.

We thank the reviewer for this information. In lines 400 ff., we correct ourselves and discuss the likelihood of SPX4 binding DNA.

8) Line 436, check spelling 'a ^{3M} mutation'

Elaborated in line 428 ff.

9) Figure 5 - move sentence beginning in line 467 to discussion - and expand on this point a little more. This aspect does not appear to be visualized.

We thank the reviewer for the suggestion. We have moved the sentence to the Discussion lines 454 ff., where we elaborate. We considered a figure illustrating topological aspects but concluded that a model would be an extrapolation too far, not least because inclusion of the α 1 helix (missing from extant models) likely has profound influence on structural models of PHR1: SPX interaction.

Nevertheless, we speculate (line 459 ff.) that differential expression of PHR1 targets might depend on P1BS repetition.

10) Throughout Materials & Methods section, state the number of replicates for assays such as protein purification, EMSA, pull-downs. Add how protein was quantified to the protein purification section.

This information is now provided in lines: 525 ff., for protein purification; 569 ff., for EMSA; 591 ff., for pull-down assay - InsP₅ matrix; and 613 ff., for pull-down assay - P1BS matrix. This also addresses Reviewer #2, Point 6.

11) Reference AGI codes for the proteins used in this study.

These are now provided in lines, 554 (SPX1); 528 ff., (ITPK4 and SHIP2).

To address Reviewer #2 (Remarks to the Author):

The manuscript NCOMMS-25-06431 investigated the binding of InsP, PP-InsP, and DNA ligands to Arabidopsis thaliana SPX1 (AtSPX1) through organic synthesis, molecular interaction experiments and molecular docking. The findings proffer a fundamentally different perspective of SPX involvement in PSR. The current manuscript only investigated the interactions of InsP, PP-InsP with DNA and AtSPX1 through molecular docking. Further experiments are needed, such as molecular dynamics simulation and cryo-electron microscopy. So I think the article should be major revised before publication.

Firstly, we thank the reviewer for consideration of our manuscript and their suggestions for its improvement. Respectfully, we have not considered cryo-electron microscopy because we consider it an unreasonable request (the approach is available to very few groups in the UK within the timescale of the revision timetable). Moreover, we would suggest that the substantive findings of our submission are not compromised by our decision not to pursue this approach.

We have, though, taken the informed suggestion of a need for molecular dynamics forwards in a serious manner. We hope the reviewer will agree that we have undertaken substantive new experiments, that we have provided a detailed description of methodology of these experiments, and that we have integrated interpretation of these experiments in our manuscript without obfuscation.

1. First, there is a lack of mass spectroscopic evidence in the PP-InsP organic synthesis section.

The organic synthesis section features in the Supplementary Information and relates to the route shown in Figure S1. Full synthetic details are provided for novel compounds, with the customary full ¹H and ³¹P spectroscopic data for all compounds and for **4** ¹³C data are provided. The starting material compound **2** is

already a known compound; the other six compounds **3-8** are new, with **8** being the target fluorescent probe. Mass spectroscopic data are indeed provided for four of the six new compounds and importantly for the target compound **8** and its immediate precursors **7** and **6**. Moreover, for **7** and **8** these are high resolution data (as indeed are provided also for **4**). The novel phosphitylating agent **3** was unstable to some degree and this may explain why we did not acquire ms data. While it is unfortunate that we are unable to provide ms spectral data for **3** and **5** we would nevertheless submit that the spectroscopic data provided more than adequately support the structural assignments of the synthetic process, steps that are in any case iterative and well-rehearsed elsewhere in the literature. What is particularly important is that characterisation of the late-stage compounds is provided, including by high resolution ms, especially for the final target itself. We would also respectfully mention that the 5-FAM-InsP₅, while providing useful support for the overall thrust of the scientific arguments of this paper is not essential to its conclusions.

2. The focus of this study is on the binding of full-length and truncated SPX1 to PHR or InsP, PP-InsP.

For clarity, the work presented in this manuscript is solely on SPX1 interaction with InsP, PP-InsP or DNA, we make no claim to have studied the binding of PHR1. We therefore limit our response to this comment to the modelling of SPX1 / InsP, PP-InsP interactions.

It is insufficient to analyze the interaction of the complex of SPX1 with PHR or InsP, PP-InsP using only static molecular docking methods. Structural analysis, or molecular dynamics simulation of the complex is required. Please modify as required.

We are pleased to describe the additional molecular dynamics simulations (MDS) requested. The methods employed are described comprehensively in Supplementary Information with heading *Molecular Dynamics Simulations and Binding Free Energy Calculations*. The results obtained therefrom are presented in Figure 1c, d and e, and in Supplementary Fig. S4a-i. In more detail, Figure 1c substitutes representative poses (i.e., that pose with the lowest average RMSD to all other conformations within the cluster) taken from the most populated clusters of conformations from MDS - of InsP₆ (left) and 1,5-InsP₈ (right), for the poses originally obtained from Schrodinger Induced Fit Docking. Figure 1d plots the estimated binding energy ΔG (requested of Point 4) obtained from the MM/PBSA (Molecular Mechanics/Poisson-Boltzmann Surface Area) method against the solution binding constants obtained from our fluorescence polarization experiments (Figure 1a, b). These results are discussed lines 128 ff., and in the legend to Supplementary Table 2.

3. The full-length proteins was used in protein purification and molecular interaction experiments, while the crystallographic structures of truncated SPX1 was employed in molecular docking of SPX1 with PHR or InsP, PP-InsP molecules.

Again, in respect of docking of SPX1 with PHR we have not studied SPX1 interaction with PHR1 and make no claim to have done so. We do discuss the literature, though.

In the study, the author mainly studies the differences between SPX1 lacking certain amino acids and full-length SPX1, which seems unreasonable and unconvincing.

Respectfully, we disagree. In respect of docking of SPX1 with InsP, PP-InsP and DNA, and as we discuss in the manuscript (lines 117 ff.), there is no full-length crystal structure of SPX1 and so it would be inaccurate to carry out MD simulations for InsPs, PP-InsPs or DNA on a model as a starting structure. Here, we comment that the truncated section is distal to the binding site, further justifying use of (SPX1 PDB: 7E40) as the receptor, line 119 ff.

The empirical solution binding constants (K_i) that we describe for a 'complete' set of inositol (pyro)phosphate ligands (Figure 1a, b) in their small (albeit, statistically not significant) variation are wholly

consistent with a literature which says that stand-alone SPX proteins show little selectivity for higher inositol (pyro)phosphate binding in solution. Our new estimations of binding energy ΔG (obtained from the suggested MM/PBSA) are now reported in Figure 1d. They broadly correlate with IC_{50} (K_i) as described lines 131 ff.

4. Regarding the molecular docking results of the interactions between SPX1 and InsP and PP-InsP, the authors did not perform bonding analysis on the key amino acids when visualizing the molecular interactions in the article, as shown in Figure S4. The binding sites of different InsP and PP-InsP with SPX1 could not be exactly the same. The binding sites were quantified by combining bonding analysis with non-bonding interactions and by analyzing the energy contribution value of each amino acid through MMPBSA. Please modify as required.

We thank the reviewer for this thoughtful intervention. By application of MDS and MM/PBSA approach we have attempted to dissect the contribution of individual amino acids in the SPX1 structure to the global binding energy of the seven different ligands characterized in Figure 1a, b. For this, we undertook in-silico alanine scanning mutagenesis. The results of this analysis are shown in Figure 1e with numerical values of the contributions of individual aa residues to $\Delta\Delta G$ also provided in Supplementary Table 1. The results are discussed lines 153 ff., and in the footnote to Supplementary Table 2. The polar contacts made by individual residues to diverse ligands are shown in Supplementary Fig. 4 (which replaces the representations obtained originally from induced fit docking).

5. Regarding the determination of affinity, the author has not provided the on-machine or fitted spectra but merely presented the numerical values of affinity.

The method employed measures polarisation of a fluorescent probe to determine IC_{50} , K_d or K_i . It does so at a single excitation wavelength and its corresponding single emission wavelength. Consequently, there are no spectra to provide. Nonetheless, Figure 1a, b provides the 4-parameter logistic (on-machine) fit of the fluorescence polarization analyses of ligand binding with the numerical values for IC_{50} . We provide the source data files.

Meanwhile, the method for ITC determination was not provided. Please modify as required.

We think we may have confused the reviewer by citing the use of ITC in other studies e.g., in original lines 346 ff., 'Figure 4a of 8'. For clarity to the reader, we have changed the terminology to 'Figure 4a of Reference 8' new line 329 and, similarly in lines 331, 426, 427 and 431.

We used fluorescence polarization to determine IC_{50} (K_i).

6. The article lacks statistical explanations.

We now provide this information.

The legend (to Figure 1) and associated narrative (line 97 ff.) describe the statistical analysis of solution ligand binding.

The legend (to Figure 2) and associated narrative (line 200 ff.) describes the statistical analysis of solution DNA binding to SPX.

The legend (to Figure 4) and associated narrative (line 279 ff.) describe the statistical analysis of the displacement of DNA-bound SPX1 by InsP and PP-InsP ligands.

The statistical significance of data (ΔG , for ligand binding) obtained from Gromacs is provided in the footnote to Table S1.

That for in silico alanine scanning mutagenesis (shown in Figure 1e and Table S2) is provided in the footnote to Table S2.

7. In addition, more information about the significance of this study should be supplemented in the discussion section of the article.

We have added in lines 302 ff. and lines 400 ff., an elaboration of how the DNA-binding function of SPX1 likely applies to other stand-alone SPX proteins e.g., SPX4, and therefore to wider transcriptional control phenomena beyond that mediated by SPX1: PHR1. Indeed, SPX4: PAP1 interaction adds organ-specific context. In our concluding paragraph, we mention the relationship of SPX function (phosphate homeostasis) to other physiological hierarchies (that of plant growth effector, auxin and jasmonate signalling). This is a topic of current interest (see reviews of Riemer et al. 2022, Ref 29; and Collins et al. 2024, Ref 34). The penultimate but one paragraph explains how the precedent of non-catalytic DNA-binding, by DNA topoisomerase 1, offers potential mechanistic insight (line 457 ff.) to stand-alone SPX function. Figure 6 (introduced, also, in answer to Reviewer #1, Point 5) integrates the DNA-binding function of SPX with InsP/PP-InsP synthesis, transport and compartmentalization.

In summary, in our revision we have shortened the abstract, we have simplified the introduction, we have retained all original references and we have described a suite of new experiments in the results section. This is presented in a revised discussion with a new model (Figure 6) that integrates inositol (pyrophosphate) synthesis and transport with SPX and PHR function in the phosphate starvation response.

Most importantly, we offer a DNA-binding function of SPX that represents a critical change of perspective in the SPX literature.

Finally, we thank N Comms for consideration of our manuscript.

Yours sincerely

Charles

Dr Charles Brearley
Professor of Biochemistry
School of Biological Sciences
University of East Anglia

Reviewer 1

The authors have addressed all of the previously raised concerns to my satisfaction. I am not a biochemist, so will rely on the second reviewer for his evaluation of the authors' response to his comments. I would say, however, that I do not think that cryo electron microscopy is required to boost the manuscript's impact. The biochemistry approach taken here is a valid characterisation of the system in its own right. Statistical information and the number of replicates has now been included. The modified Figure 6 adds a lot of value to the discussion of the findings.

Just one minor concern:

In Figure 6, please remove the SPX1-PHR1 module in the cytosol. While it has been shown that SPX4-PHR1 interact in the cytosol – and can also shuttle to the nucleus (either together or individually), SPX1 (SPX2)-PHR1 interaction has so far only been observed in the nucleus.

We have removed this and the new figure 6 has been inserted. We thank the reviewer for noticing this error.

Reviewer 2

The manuscript NCOMMS-25-06431 investigated the binding of InsP, PP-InsP, and DNA ligands to *Arabidopsis thaliana* SPX1 (AtSPX1) through organic synthesis, molecular interaction experiments and molecular docking. Major revisions are still needed before publication.

1. First of all, the introduction section does not clearly explain the reasons for choosing the SPX1 protein or the previous shortcomings. The significance of this study should be emphasized in the introduction section. And the innovative aspects of this article need to be more prominently highlighted.

We have added some sentences to the introduction to make this clearer.

2. The supplementary kinetic simulation in the article only simulated for 10 nanoseconds, which is completely unconvincing. It should have simulated for more than 100 ns.

With respect, it is generally accepted that in order to obtain better predictions of binding free energies from mm/PBSA calculations longer MD simulations are not necessarily beneficial. Numerous studies have concluded that the length of the simulations is not of critical importance to the accuracy of the calculations as it seems that the impact of the MD simulation length on free energy calculations is system-dependent. Indeed, simulation lengths of less than 5 ns are considered to be reasonable¹ and consequently we decided to calculate 10 ns trajectories, of which the first 5 ns concluded equilibration. We used the final 5 ns of these, consisting of 100 individual frames separated by 50 ps, to calculate binding energies using the gmx_mmPBSA method.

However, this comment made by the reviewer led us to reflect on our method. It has been demonstrated that it is more effective to run a number of short independent simulations than a single long one^{2,3} as the latter will underestimate the uncertainty in the result. Basically, multiple short simulations improve the sampling performance. Hence, we decided to adopt the standard method of employing random number seeds to generate different starting velocities for a further nine independent MD simulations for each complex, giving 10 replica trajectories and a total sampling time of 50 ns per ligand. This number of replica repeat simulations follows the recommendations of Knapp et al.⁴. Having ascertained that the individual simulations had equilibrated we employed the same method as previously, extracting from each 5 ns trajectory 100 individual frames separated by 50 ps to calculate replica trajectory-averaged binding energies using the gmx_mmPBSA method.

The resulting rank order of ligands sorted by estimated binding free energy calculated using this multi-trajectory approach differs marginally from that observed from a single trajectory. However, the resulting standard deviations in the binding free energies are somewhat larger than those calculated previously from a single trajectory and so it is the case that more independent trajectories will be needed to reduce the uncertainty. However, we have chosen not to do this because of the limited time available to perform the additional calculations and because we are reassured that the qualitative agreement between predicted binding energies and IC₅₀ values obtained from the single trajectory approach was maintained over the multiple independent trajectory approach. We would like to emphasize at this point that our intention has not been to generate highly accurate binding free energy estimates for the ligands but rather to help validate the *in vitro* displacement assays and to give insights into how the ligands might bind to SPX1 and the receptor residues involved. We feel that our latest results achieve this goal.

1. Hou, T.; Wang, J.; Li, Y.; Wang, W. Assessing the Performance of the MM/PBSA and MM/GBSA Methods: I. The Accuracy of Binding Free Energy Calculations Based on Molecular Dynamics Simulations. *J. Chem. Inf. Model.* 2011, 51, 69–82.
 2. Virtanen, S. I.; Niinivehmas, S. P.; Pentikäinen, O. T. Case Specific Performance of MM-PBSA, MM-GBSA, and SIE in Virtual Screening. *J. Mol. Graphics Model.* 2015, 62, 303–318.
 3. Wang EC, Sun HY, Wang JM, Wang Z, Liu H, Zhang JZH, Hou TG. From End-Point Binding Free Energy Calculation with MM/PBSA and MM/GBSA: Strategies and Applications in Drug Design. *Chem. Rev.* 2019, 119(16), 9478–9508.
 4. Knapp B, Ospina L, Deane C<M. Avoiding False Positive Conclusions in Molecular Simulation: The Importance of Replicas. *J. Chem. Theory Comput.* 2018, 14, 12, 6127–6138
- 3. The kinetic simulation merely analyzed the results of MMPBSA. RMSD, RMSF, SASA, hydrogen bonds and distance evolution are important criteria for evaluating the reliability of the simulation. Please provide a diagram to supplement.**

We are happy to provide figures S6-S12 showing the evolution of RMSD, RMSF, SASA and number of ion pairs found over the course of the production phase of the simulations for each ligand. Separate plots of the RMSD for atoms of the ligands and binding site residues are presented as figure S13. These provide justification for our conclusion that the simulation systems had equilibrated within the first 5 ns of the trajectories. Unfortunately, the reviewer did not state which distance(s) should be monitored and so we

have been unable to satisfy this request.

- 4. The author points out that Inositol (pyro)phosphates and DNA are reciprocally competing ligands of SPX1. The molecular dynamics simulation did not simulate DNA and SPX1. It is necessary to provide evidence to prove this phenomenon.**

We feel the message of the study, that SPX1 binds inositol pyrophosphates and DNA reciprocally, is supported by our numerous in vitro studies where DNA and inositol pyrophosphates displace each other in EMSA and polarization studies. Given we are not claiming to have determined the site, sequence or orientation of DNA binding, modelling is unnecessary. Modelling of proteins with DNA is notoriously unpredictable and inaccurate, particularly for lower affinity binding as may be expected here. To begin to model the protein-DNA interaction, identification of the binding surface and DNA sequence is needed. Without this information, modelling would be more likely to produce a result that could mislead readers and so we choose not to carry this work out. Given that both the DNA binding surface and the preferred binding sequence of SPX1 is unknown, to complete this work to an acceptable accuracy would take months to years to carry out and is beyond the scope of this paper. We have demonstrated in vitro the binding of DNA to the SPX1 protein using multiple methods. We have also used the predictive binding site server RoseTTAFoldDNA (shown in figure 3c) to predict DNA binding position showing that it is feasible that they may overlap. We propose this as a model in our discussion, but are clear that this is only a proposed model.

- 5. The method for virtual scanning of mutations needs to be supplemented. The header in Table S2 states that it presents the results of scanning mutations, but it also says, "Gbind values, calculated over 100 frames (5 ns) of the molecular-dynamics trajectories, are reported with standard deviation (in units of kJ/mol)." Could you please clarify whether this is the result of scanning mutations or the result of MMPBSA energy decomposition? If it is the result of scanning mutations, could you explain how the deviation values were obtained?**

The results of our virtual scanning of mutations in the SPX1 ligand binding pocket shown in Figure S4b and Table S2 have been updated using the results of all 10 independent simulations for each ligand. Table S2 reports the results of scanning alanine mutagenesis of SPX1 ligand-binding site residues and did not result from MMPBSA energy decomposition. The standard deviations shown result from the distribution of free energy values calculated for each the 100 frames at 50 ps intervals between timepoints 5 and 10 ns of each of 10 independent trajectories for each ligand (i.e. 1000 frames in total). We have added some further explanatory text to the table to clarify the methodology used.

- 6. The visualizations in Figures 1c and S4 of the article do not specify which specific conformation the simulation was carried out at. Therefore, a FEL analysis should be conducted on the simulation trajectories to identify the most stable conformation for analysis. The most important point is that based on the visualized graph, neither amino acids nor small molecules' hydrogen were detected. It is very likely that before the simulation, the hydrogen addition treatment for small molecules and proteins was not carried out. Therefore, the results obtained in this way are incorrect.**

As is stated in the Protein Modelling section of the Supplementary Information, simulations were initiated

from the lowest energy pose for each ligand arrived at by Induced Fit Docking (IFD) performed using the Maestro-Schrödinger suite. In the latest version of the manuscript we have chosen to present a representative bound state for each ligand as that found to be the middle structure of the most populous cluster found over the combined 10 independent simulations i.e. that pose with the lowest average RMSD to all other structures within the cluster. The Amber99SB-ILDN force field was employed for molecular dynamics simulations (see Supplementary Information section 'Molecular Dynamics Simulations and Binding Free Energy Calculations'). This is an all-atom force field and hydrogen atoms were simply removed from figures 1c and S5 for clarity. Additional explanatory text has been added to the legends of Figures 1 and S5 to this end.

As suggested by the reviewer, we have carried out Free Energy Landscape calculations for each of the ligands bound to OsSPX1 and for the uncomplexed protein and present the results in the Supplementary Information as Figure S14. These plots suggest that InsP₆ and diphosphoinositol phosphate binding to SPX1 can alter the principal components (PC1 and PC2) of the system identified by molecular dynamics simulations i.e. those corresponding to low frequency molecular motions. This is unsurprising and other authors have noted that ligand binding can remodel the conformational space of a protein, shifting energy minima and stabilizing specific states such as open or closed conformations⁵. However, we chose not to identify the most stable bound conformation for each ligand via FEL analysis as suggested by the reviewer. This is a perfectly reasonable suggestion but we believe that our multi-trajectory approach leads to an improved sampling of conformational space and therefore perhaps a more realistic approach to estimation of qualitative free energies of binding.

5. Messina TC, Talaga DS. Protein free energy landscapes remodeled by ligand binding. *Biophys J.* 2007 Jul 15;93(2):579-85. doi: 10.1529/biophysj.107.103911.

7. Figure S4 should have its trajectory analysis improved. Analyze the interaction graphs for different time periods, such as 0 ns, 25 ns, 50 ns, 75 ns, and 100 ns.

Analysis of our trajectories revealed that equilibration had been achieved by 5ns and so analysis of the SPX1-ligand interactions would be inappropriate at any other time points. As discussed above, we have chosen not to generate a single long trajectory but rather 10 short independent trajectories for each complex. We believe that the most populous state of each ligand as bound to SPX1 found over the 50 ns of the combined 10 independent trajectories and shown in an updated version of Figure S4 (now Figure S5) provides useful insights into the interactions made with binding site residues.

8. The ITC results in the article only reproduced 8 data points, and no obvious S-shaped pattern was observed as shown in Figure 2B. The results are unreliable. Additionally, please provide detailed measurement methods. At the same time, please attach the original image of the machine operation as an attachment.

Again, as in our previous response to reviews, we stress that we did not carry out ITC. The results you refer to are fluorescence polarisation assays, and we believe that the figure legend and label on the graphs are clear to state this is anisotropy, not ITC. Therefore we cannot cover any of the points the reviewer asks for.

9. Figures 5 and 6 are both schematic diagrams. It seems unnecessary.

Another reviewer requested figure 6 and we believe it summarises current knowledge well.

10. Many of the icons in the article do not have sufficient clarity and their quality has not reached the publication standards.

We are not clear what the reviewer means by “icons”, are they referring to the figures? Assuming so, with regard to quality of images, they are clear in our version of the manuscript. If the manuscript is accepted for publication we have individual high quality images for each figure to upload, rather than them being embedded inside a word or PDF document as was the case for initial submission. We will of course work with the editor to ensure high quality figures are available to readers and have never had a problem providing this in our previous publications.

11. First, there is a lack of mass spectroscopic evidence in the PP-InsP organic synthesis section.

We repeat our comments as per last round reviewer 2 comments. Please see the supplementary information section for the mass spectroscopic evidence as described below:

The organic synthesis section features in the SI and relates to the route shown in Figure S1. Full synthetic details are provided for novel compounds, with the customary full ^1H and ^{31}P spectroscopic data for all compounds and for 4 ^{13}C data are provided. The starting material compound 2 is already a known compound; the other six compounds 3-8 are new, with 8 being the target fluorescent probe. Mass spectroscopic data are indeed provided for four of the six new compounds and importantly for the target compound 8 and its immediate precursors 7 and 6. Moreover, for 7 and 8 these are high resolution data (as indeed are provided also for 4). The novel phosphitylating agent 3 was unstable to some degree and this may explain why we did not acquire ms data. While it is unfortunate that we are unable to provide ms spectral data for 3 and 5 we would nevertheless submit that the spectroscopic data actually provided more than adequately support the structural assignments of the synthetic process, steps that are in any case iterative and well-rehearsed elsewhere in the literature. What is particularly important is that characterisation of the late stage compounds is provided, including by high resolution ms, especially for the final target itself.

Finally, we thank our reviewers and editor for the opportunity to submit a revised manuscript.

Yours sincerely

Charles Brearley